# Constrained Optimization to Train Neural Networks on Critical and Under-Represented Classes

**Sara Sangalli**[1], **Ertunc Erdil**[1], **Andreas Hoetker**[2], **Olivio Donati**[2], **Ender Konukoglu**[1]

[1] Computer Vision Lab, ETH Zürich

[2] Institute for Diagnostic and Interventional Radiology, Universitätsspital Zürich

`sara.sangalli@vision.ee.ethz.ch`

## Abstract

Deep neural networks (DNNs) are notorious for making more mistakes for the classes that have substantially fewer samples than the others during training. Such class imbalance is ubiquitous in clinical applications and very crucial to handle because the classes with fewer samples most often correspond to critical cases (e.g., cancer) where misclassifications can have severe consequences. Not to miss such cases, binary classifiers need to be operated at high True Positive Rates (TPRs) by setting a higher threshold, but this comes at the cost of very high False Positive Rates (FPRs) for problems with class imbalance. Existing methods for learning under class imbalance most often do not take this into account. We argue that prediction accuracy should be improved by emphasizing the reduction of FPRs at high TPRs for problems where misclassification of the positive, i.e. critical, class samples are associated with higher cost. To this end, we pose the training of a DNN for binary classification as a constrained optimization problem and introduce a novel constraint that can be used with existing loss functions to enforce maximal area under the ROC curve (AUC) through prioritizing FPR reduction at high TPR. We solve the resulting constrained optimization problem using an Augmented Lagrangian method (ALM). Going beyond binary, we also propose two possible extensions of the proposed constraint for multi-class classification problems. We present experimental results for image-based binary and multi-class classification applications using an in-house medical imaging dataset, CIFAR10, and CIFAR100. Our results demonstrate that the proposed method improves the baselines in majority of the cases by attaining higher accuracy on critical classes while reducing the misclassification rate for the non-critical class samples.[1]

## 1 Introduction

Deep Neural Networks (DNNs) perform extremely well in many classification tasks when sufficiently large and representative datasets are available for training. However, in many real world applications, it is not uncommon to encounter highly-skewed class distributions, i.e., majority of the data belong to only a few classes while some classes are represented with scarce instances. Training DNNs on such imbalanced datasets leads to models that are biased toward majority classes with poor prediction accuracy for samples of minority class. While this is problematic for all such applications, it poses an even greater issue for "critical" applications where misclassifying samples belonging to the minority class can have severe consequences. One domain where such applications are common and machine learning is having an important impact is medical imaging.

In medical imaging, applications with data imbalance are ubiquitous [14] and costs of making some types of mistakes are more severe than others. For instance, in a diagnosis application, discarding a

---

[1]Code is available at: `https://github.com/salusanga/alm-dnn`.

cancer case as healthy (False Negative) is more costly than classifying a healthy subject as having cancer (False Positive). While the latter creates burden for the subject as well as health-care system through additional tests that may be invasive and expensive, the former, i.e. failure to identify a cancerous case, would delay the diagnosis and jeopardise the treatment success. In such applications, binary classifiers are operated at high True Positive Rates (TPRs) even when this means having higher False Positive Rates (FPRs). To make matters more complicated, there are usually significantly fewer samples to represent critical classes, where mistakes are more severe. For instance, in [6] authors found out that in prostate cancer screening only 30% of even the most suspicious cases identified with initial testing actually have cancer. Such class imbalance increases the FPR even higher in "critical" applications, because the models tend to misclassify minority classes more often. Useful algorithms need to achieve low FPR at high TPR operating points, even under class imbalance.

While various methods for learning with imbalanced datasets exist, to the best of our knowledge, these methods do not take into account the fact that "critical" applications need to be operated at high accuracy for the critical classes. We believe that for such applications ensuring low misclassification rate for the non-critical samples and high accuracy for the critical classes should be the main goal, to make binary classifiers useful in practice. This motivates us to design new strategies for training DNNs for classification.

**Contribution:** In this paper, we pose the training of a DNN for binary classification under class imbalance as a constrained optimization problem and propose a novel constraint that can be used with existing loss functions. We define the constraint using Mann-Whitney statistics [16] in order to maximize the AUC, but in an asymmetric way to favor reduction of false positives at high true positive (or low false negative) rates. Then, we transfer the constrained problem to its dual unconstrained optimization problem using an Augmented Lagrangian method (ALM) [2]. We optimize the resulting loss function using stochastic gradient descent. Unlike the existing methods that directly optimize AUC, we incorporate AUC optimization in a principled way into a constrained optimization framework. We finally present two possible extensions of the proposed constraint for multi-class classification problems.

We present an extensive evaluation of the proposed method for image-based binary and multi-class classification problems on three datasets: an in-house medical dataset for prostate cancer, CIFAR10, and CIFAR100 [11]. In all datasets, we perform experiments by simulating different class imbalance ratios. In our experiments, we apply the proposed constraint to 9 different baseline loss functions, most of which were proposed to handle class imbalance. We compare the results with the baselines without any constraint. The results demonstrate that the proposed method improves the baselines in majority of the cases.

## 2 Related work

Various methods have already been proposed to learn better models with class-imbalanced datasets. We group the existing methods into three categories: cost sensitive training-based methods, sampling-based and classifier-based methods. Here, we focus on the first one and present related work for the other groups in the supplementary material for space reasons.

**Cost sensitive training-based methods:** This family of methods aims at handling class imbalance by designing an appropriate loss function to be used during training. In particular, they design loss functions to give more emphasis to the samples belonging to minority class, or the class with higher associated risk, than the majority ones during training [22]. [25] proposes a loss function, which we refer to as Weighted BCE (W-BCE), where samples from minority class are multiplied by a constant to introduce more cost to misclassification of those samples. [10] proposes a function that aims to learn more discriminative latent representations by enforcing DNNs to maintain inter-cluster and inter-class margins, where clusters are formed using k-means clustering. They demonstrate that the tighter constraint inherently reduces class imbalance. In a more recent work, [5] proposes a loss function called class-balanced binary cross-entropy (CB-BCE) to weight BCE inversely proportionally to the class frequencies to amplify the loss for samples from the minority class. In a similar vein, [13] modifies BCE and propose symmetric focal loss (S-FL) by multiplying it with the inverse of the prediction probability to introduce more cost to the samples that DNNs are not very confident. [15] introduces symmetric margin loss (S-ML) by introducing a margin to the BCE loss. [12] investigates different loss functions such as S-FL and S-ML, and propose their asymmetric versions, A-FL and

A-ML, by introducing a margin for samples from the minority class to handle class imbalance. In a different line of work, [23] proposes a method called mean squared false error by performing simple yet effective modification to the mean squared error (MSE) loss. Unlike MSE, which computes an average error from all samples without considering their classes, this loss computes a mean error for each class and averages them. [3] proposes a label-distribution-aware margin (LDAM) loss motivated by minimizing a margin-based generalization bound, optionally coupled with a training schedule that defers re-weighting until after the initial stage. [19] introduces balanced meta-softmax for long-tailed recognition, which accommodates the label distribution shift between training and testing, as well as a meta sampler that learns to re-sample training set by meta-learning. [21] proposes a loss that ignores the gradient from samples of large classes for the rare ones, making the training more fair.

A particular group within the cost sensitive training-based methods focuses on optimizing AUC and our method falls into this group. AUC optimization is an ideal choice for class imbalance since AUC is not sensitive to class distributions [4]. [18] proposes a support vector machine (SVM) based loss function that maximizes AUC and demonstrates its effectiveness for the class imbalance. [26] approaches the class imbalance problem from online learning perspective and proposes an AUC optimization-based loss function. [7] proposes a one-pass method for AUC optimization that does not require storing data unlike the previous online methods. Another online AUC optimization method proposed by [24] formulates AUC optimization as a convex-concave saddle point problem. Despite their usefulness, all aforementioned AUC optimization-based methods were applied to linear predictive models, as this allows to simplify the Mann-Whitney statistics [16] for the definition of AUC, and their performance on DNNs is unknown. [20] applies online AUC optimization on a small dataset for breast cancer detection where they also mention that extension to larger datasets may not be feasible. In a very recent work called mini-batch AUC (MBAUC) [8], authors extend AUC optimization to non-linear models with DNNs by optimizing AUC with mini-batches and demonstrate its effectiveness on various datasets.

The proposed constrained optimization method differs from the existing works in that it enforces maximal AUC as a constraint in a way that favors reducing FPR at high TPR and can be used with existing loss functions.

## 3    Background - Augmented Lagrangian method (ALM)

A generic optimization problem for an objective function $F(\theta)$ subject to the constraints $\mathcal{C}(\theta) = \{c_1(\theta), ..., c_m(\theta)\}$ can be expressed as [2, 17]:

$$\arg\min_{\theta \in \Theta} F(\theta); \quad \text{subject to} \ \ \mathcal{C}(\theta) \tag{1}$$

Augmented Lagrangian method (ALM) [1], also known as methods of multipliers, converts the constrained optimization problem in Eq. (1) to an unconstrained optimization problem. ALM is proposed to overcome the limitations of two earlier methods called quadratic penalty method and method of Lagrangian multipliers which suffer from training instability and non-convergence due to the difficulty of convexifying loss functions[2]. In ALM, the penalty concept is merged with the primal-dual philosophy of classic Lagrangian function. In such methods, the penalty term is added not to the objective function $F(\theta)$ but rather to its Lagrangian function, thus forming the Augmented Lagrangian Function:

$$\mathcal{L}_\mu(\theta, \lambda) = F(\theta) + \mu \sum_{i=1}^{m} \big\| c_i(\theta) \big\|^2 + \sum_{i=1}^{m} \lambda_i c_i(\theta) \tag{2}$$

In practice, this method consists in iteratively solving a sequence of problems as:

$$\max_{\lambda^k} \min_{\theta} \mathcal{L}_{\mu^k}(\theta, \lambda^k), \quad \theta \in \Theta \tag{3}$$

Where $\{\lambda^k\}$ is a bounded sequence in $\mathcal{R}^m$, updated as $\lambda_i^{k+1} = \lambda_i^k + \mu c_i(\theta)$. $\{\mu^k\}$ is a positive penalty parameter sequence, with $0 < \mu^k \leq \mu^{k+1}$, $\mu^k \to \infty$, which may be either pre-selected or generated during the computation according to a defined scheme. In ALM, increasing $\mu^k$ indefinitely

---

[2]Please consult supplementary material for more details of quadratic penalty method and method of Lagrangian multipliers.

is not necessary as in the quadratic penalty method. Thus, it does not suffer from training instabilities due to the constraint prevailing $F(\theta)$. Furthermore, it does not require convexity assumption as in the method of Lagrange multipliers to ensure convergence [1].

## 4 Proposed methods

### 4.1 Proposed constraint for binary classification

Let $F(\theta)$ be a generic loss function that is used to train classification DNNs, $f_\theta(.)$, for binary problems. Let us also define $p \triangleq \{x_1^p, \cdots, x_{|p|}^p\}$ and $n \triangleq \{x_1^n, \cdots, x_{|n|}^n\}$ as the sets of samples of the positive (critical) and negative classes, respectively. Note that we choose $p$ as the minority class in our description, i.e., $|p| < |n|$, and assume that this is a critical class associated with higher risk of making a mistake. We define our constrained optimization problem as follows:

$$\underset{\theta}{\arg\min} \ F(\theta)$$

$$\text{subject to } \sum_{k=1}^{|n|} \max\left(0, -\left(f_\theta(x_j^p) - f_\theta(x_k^n)\right) + \delta\right) = 0, \ \ j \in \{1, ..., |p|\}, \tag{4}$$

where $f_\theta(x)$ indicates output probability of the DNN on input $x$. Note that the constraint states that the output of the DNN for each sample of critical class should be larger than the outputs of all of the negative samples by a margin $\delta$. Satisfying the constraint would directly ensure maximal AUC [16].

We define the equivalent unconstrained version of Eq. (4) by writing it in the form given in Eq. (2)

$$\mathcal{L}_\mu(\theta, \lambda) = F(\theta) + \frac{\mu \sum_{j=1}^{|p|} q_j^2}{2 \cdot |p| \cdot |n|} + \frac{\sum_{j=1}^{|p|} \lambda_j \cdot q_j}{|p| \cdot |n|} \tag{5}$$

where, $q_j = \sum_{k=1}^{|n|} \max(0, -(f_\theta(x_j^p) - f_\theta(x_k^n)) + \delta)$, $\mu$ is the penalty coefficient corresponding to the quadratic penalty term, $\lambda_j$ is the estimate of Lagrange multiplier corresponding to each positive training sample $j$, and $\delta$ is the margin that we determine using a validation dataset.

The crucial aspect of the formulation is the asymmetry between positive and negative classes. A constraint is defined for each sample from the positive class, thus each positive sample gets a separate Lagrange multiplier. This form prioritizes the reduction of FPR at high TPR values as illustrated next.

We use Algorithm 1 to estimate the parameters of a DNN, $\theta$, using the proposed loss function in Eq. 5. The parameters $\theta$ are updated with every batch using gradient descent with learning rate $\alpha$. Concurrently, $\mu$ is increased using a multiplicative coefficient $\rho$ only when a chosen metric on validation is not improved, by a margin to avoid training instabilities. The validation metric (*ValMETRIC*) is: 1) Validation AUC for the binary setup; 2) Validation Accuracy for the multi-class version. We update $\lambda_j$ in each iteration for each positive sample $x_j^p$.

---

**Algorithm 1** ALM for Training DNNs

---
    **Input:** $\theta^{(0)}, \mu^{(0)}, \lambda_j^{(0)}, \rho$;
    **for** $t = 1, \ldots, T$ **do**
        **for** each mini-batch of $x_B$ with size $B$ **do**
            $y_B = f(X_B)$;
            Calculate $q_j^{(t)}$;                               $\triangleright \forall j \in [1, B]$ and $y_j = y^+$
            $\theta^{(t+1)} \leftarrow \theta^{(t)} - \alpha \cdot \nabla_\theta \mathcal{L}_\mu(\theta^{(t)}, \lambda)$;
            $\lambda_j^{(t+1)} \leftarrow \lambda_j^{(t)} + \mu^{(t)} \cdot q_j^{(t)}$;                 $\triangleright \forall j \in [1, B]$ and $y_j = y^+$
        **end for**
        **if** $ValMETRIC^{(t)} < ValMETRIC^{(t-1)}$ **then**
            $\mu^{(t+1)} \leftarrow \mu^{(t)} \cdot \rho$;
        **else**
            $\mu^{(t+1)} \leftarrow \mu^{(t)}$;
        **end if**
    **end for**
    Return $\theta^{(t+1)}$

---

## 4.2  Toy example describing our design choice

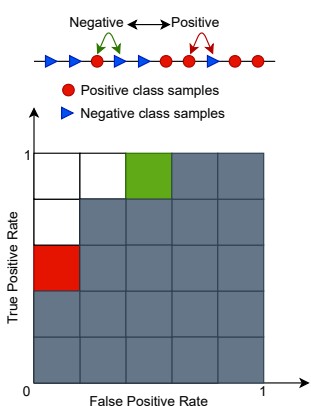

Figure 1: Toy example that illustrates different optimizations, which yield the same improvement in AUC. The one that adds the green box to the AUC however, leads to lower FPRs at the highest TPRs. The ALM given in Eq. (5) prefers adding the green box to improve the AUC rather than the red one.

Recall that our goal with the constraint and the final augmented loss function is to maximize AUC through minimizing FPR for high TPR. The design of the constraint is crucial to achieve this goal. In Fig. 1, we demonstrate this on a toy example. At the top we show 10 data samples and order them with respect to a classifier's output for the samples, i.e., samples on the right are assumed to yield higher output than those on the left. The gray area in the figure below shows the AUC for the toy data samples. Consider two different optimizations to increase the AUC. One adds the red box and the other adds the green box to the gray area. Both optimizations lead to exactly the same AUC improvement, however, only adding the green box reduces FPRs at the highest TPRs.

In the proposed method, we design the constraint to achieve lower FPR at high TPR, such that the optimization can reduce cost more by adding the green box instead of the red box. To see this, let us assume that the distances between all the successive markers in the figure are the same, and we denote this by $\Delta$. Note that the distances between markers indicate the differences between the outputs of the classifiers for the samples corresponding to the markers. Let us also further assume that all the Lagrange multipliers have the same value. In this case, one can verify that if the optimization swaps the locations of the left most positive sample (red circle marker) and the negative sample to its immediate right (blue triangle marker), the cost due to the constraint in Eq. (5) decreases by $\frac{39\mu\Delta^2}{50} + \frac{3\lambda\Delta}{25}$. Swapping the locations of the right most negative sample and the positive sample to its left decreases the cost due to the constraint by $\frac{19\mu\Delta^2}{50} + \frac{3\lambda\Delta}{25}$ [3]. So, from the cost perspective, the optimization should prefer the former swap over the latter, which corresponds to adding the green box to the AUC instead of red box. Therefore, the augmented Lagrangian cost would be decreased further when FPR at the highest TPR is reduced rather than increasing the TPR at the lowest FPR. Instead of defining the constraint for each positive sample in Eq. (5), if we were to define it for each negative sample in the exactly opposite way, i.e., $\sum_{j=1}^{|p|} \max(0, -(f_\theta(x_j^p) - f_\theta(x_k^n)) + \delta) = 0, \ k \in \{1, \ldots, |n|\}$, then the situation would be reversed. The optimization would prefer adding the red box over the green box to decrease the cost further. If we were to define a constraint for each positive-negative pair, i.e., $f_\theta(x_j^p) > f_\theta(x_k^n), \forall j$ and $k$, then adding the red or the green box to improve the AUC would yield exactly the same decrease in the cost.

## 4.3  Extensions to multi-class classification

The proposed constraint can also be extended to multi-class classification with slight modifications. Let us assume that we have a multi-class classification problem with $C$ classes. In this case, there can be multiple critical and non-critical classes depending on the application. Let us define the corresponding family of sets, i.e., sets of sets, as $P = \{p^1, \cdots, p^{|P|}\}$ and $N = \{n^1, \cdots, n^{|N|}\}$ where the sets $p^i$ and $n^i$ are as defined in the previous section. Also, note that $|P \cup N| = C$ and $P \cap N = \emptyset$.

A main difference between binary and multi-class classification is the dimension of the output. While $f_\theta(x)$ was a single value for binary classification, in multi-class problem it is a vector with one value for each class. Using the notation for the positive and negative classes, we write the output of the

---

[3]Derivations for this toy example and theoretical insights for our design choice are provided in the supplementary material for convenience.

network as $f_\theta(x) = \{f_\theta^{p^1}, \cdots, f_\theta^{p^{|P|}}, f_\theta^{n^1}, \cdots, f_\theta^{n^{|N|}}\}$. Based on this, we define our first constraint for multi-class classification as

$$q_{cj} = \sum_{i=1}^{|N|} \sum_{k=1}^{|n^i|} \max\left(0, -\left(f_\theta^{p^c}(x_j^{p^c}) - f_\theta^{p^c}(x_k^{n^i})\right) + \delta\right), \ \ c \in \{1, \cdots, |P|\}, \ j \in \{1, ..., p^c\} \quad (6)$$

where $f_\theta^{p^c}(x)$ indicates the output probability of DNN for the critical class $p^c$. Note that the constraint in Eq. (6) enforces that output probabilities of the DNN for critical classes should be larger for critical samples than for samples of other classes. We refer to ALM with the first constraint as $\text{ALM}_{m,1}$.

An alternative definition of constraint is also possible. The first constraint enforces that probabilities of the critical classes should be larger for samples of each critical class. We can go a step further and add penalty to enforce that probabilities of the non-critical classes should be smaller for critical samples than the non-critical samples belonging to the corresponding class. We denote ALM with the second constraint as $\text{ALM}_{m,2}$.

$$\begin{aligned}
q_{cj} = \sum_{i=1}^{|N|} \sum_{k=1}^{|n^i|} &\max\left(0, -\left(f_\theta^{p^c}(x_j^{p^c}) - f_\theta^{p^c}(x_k^{n^i})\right) + \delta\right) \\
&+ \max\left(0, \left(f_\theta^{n^i}(x_j^{p^c}) - f_\theta^{n^i}(x_k^{n^i})\right) + \delta\right), \ \ c \in \{1, \cdots, |P|\}, \ j \in \{1, ..., p^c\}
\end{aligned} \quad (7)$$

In this formulation, the penalty does not view all non-critical classes as one, but also contributes to improving classification accuracy in those classes as well, by increasing the gap between the corresponding non-critical probability of critical and non-critical samples.

## 5 Experiments

In this section, we present our experimental evaluations on image-based classification tasks. We perform experiments on three datasets: an in-house MRI medical dataset for prostate cancer and two publicly available computer vision datasets, CIFAR10 and CIFAR100 [11].

In our evaluation, we experiment with different existing loss functions, most of which have been designed to handle class imbalance: classic binary and multi-class cross-entropy (BCE, CE), symmetric margin loss (S-ML) [15], symmetric focal loss (S-FL) [13], asymmetric margin loss (A-ML) and focal loss (A-FL) [12], cost-weighted BCE (WBCE) [25], class-balanced BCE (CB-BCE) and CE (CB-CE) [5], label-distribution-aware margin loss (LDAM) [3]. We first train DNNs for classification using only the loss functions and then using our method, which adds the proposed constraint to the loss function and solves Eq. (5), and we compare the classification performances. In addition, we also compare the proposed method with directly optimizing AUC using the mini-batch AUC (MBAUC) method proposed in [8] for the binary case.

The proposed method is implemented in PyTorch and we run all experiments on a Nvidia GeForce GTX Titan X GPU with 12GB memory.

### 5.1 Datasets

**Prostate MRI dataset** consists of 2 distinct cohorts: 1) a group of 300 multiparametric prostate MRI studies used for training and 2) another group of 100 multiparametric prostate MRI studies used for testing the trained DNNs. There is no overlap between the groups. Consent from the each subject is obtained to use the data for research purposes. Two board-certified radiologists with 10 and 7 years of experience in dedicated prostate imaging independently reviewed all examinations of the training set and test set and scored whether Dynamic Contrast Enhanced (DCE) sequences would be beneficial for cancer diagnosis. After completion of readings, a consensus was reached by the two readers by reviewing all examinations with discrepant decisions. The goal of the binary classification here is to identify subjects who do not require additional DCE imaging for accurate diagnosis, so they can be spared from unnecessary injection and cost and duration of the scanning can be reduced. In our experiments, we randomly split 20% of the training cohort as validation set by keeping the class imbalance consistent across the datasets. In both training and testing cohorts, positive samples represent 13% of all the patients which leads to an inherent 1:8 class ratio.

**CIFAR10 and CIFAR100 datasets** For CIFAR10, in the binary experiments we randomly select 2 classes among 10 to pose a binary classification problem. Additional experiments with different random selections are in the supplementary material. We use all the available training samples of the selected majority class, while we randomly pick a varying number of training samples for the minority class to obtain different class ratios, up to 1:200. We select 100 samples per class as validation set to determine hyper-parameters. For testing, we use all the available test samples, which consist of 1000 images for each class. For CIFAR100 in the binary setup we select one super-class as majority one, and a sub-class of another super-category as minority one. For training we use all the available 2250 samples of the selected larger class, and we randomly pick a varying number of training samples for the minority class to obtain different class ratios, up to 1:200. We use 50 samples for each original sub-class as validation set to perform the parameter search. For testing, we use all the available test samples, which consist of 100 images for each sub-class. For the multi-class experiments, long-tailed versions of CIFAR10 and CIFAR100 are built accordingly to [5, 3] and we consider the smallest class as the only critical class[4]. To be consistent with recent works [5, 3, 19], we use balanced validation and test sets in all the experiments on CIFAR datasets.

## 5.2 Training details

**Network architectures:** For the prostate MRI dataset, we use a 3D CNN that consist of cascaded 3D convolution, 3D max-pooling, intermediate ReLu activation functions and Sigmoid in the final output. For the binary CIFAR10 and CIFAR100 datasets, we use ResNet-10, while ResNet-32 [9] is adopted for the multi-class experiments, consistently with [3, 5]. Further training details can be found in the supplementary material.

**Ensembling for higher reliability:** Model reliability is very crucial when training DNNs. Dealing with small datasets may lead to dataset-dependent results even with the random splits, which could completely hinder objective evaluation. To weaken this phenomenon, we adopt the following ensembling strategy on MRI and for consistency we apply it on the binary CIFAR10 and CIFAR100 as well. Given a dataset and a class ratio, we create 10 random stratified splits of the dataset and train 10 models independently. The larger portions are used for training and the smaller portions for choosing hyper-parameters. During inference, all the models are applied on test samples and predictions are averaged in the logit space before the sigmoid function to yield the final prediction. We apply the ensembling to all the binary models we experiment with. This practice attenuates data dependency and we observed that the final AUC is improved when compared to the average of AUCs of different models, as presented in Table 3 for MRI sequences and in supplementary material for CIFAR10 and CIFAR100.

**Hyper-parameters selection:** Selection of the best hyper-parameters is crucial both to ensure proper and fair evaluation of the methods and to understand the true performance of any model. To achieve this, we perform grid-search to determine the hyper-parameters that yield the highest AUC for the binary experiments. For the multi-class tests we select the model that achieved the best overall accuracy on the validation set, in order to be consistent with the related works. The test sets in all experiments are not used for hyper-parameter selection. To reduce computational load, we select the common hyper-parameters such as the optimizer, learning rate, and the activation functions in the DNNs based on their performance with BCE loss function for the binary experiments, and consistently with [3] for the multi-class ones. Then, we keep them fixed in all experiments on the same dataset.

Besides the common hyper-parameters, the majority of the existing methods have hyper-parameters that crucially affect their performance. Namely, these hyper-parameters are margin $m$ for S-ML and A-ML, exponent $\gamma$ for S-FL and A-FL, weight of the cost $c$ for WBCE and $\beta$ for CB-BCE. We select best values for these hyper-parameters from the respective candidate sets that we create based on the information provided in the original papers for each of them.

In the proposed method, there are 4 parameters to be set: $\mu^{(0)}$, $\lambda^{(0)}$, $\rho$, and $\delta$. Thus, hyperparameters' search is an important aspect of the proposed method. $\mu^{(0)}$, $\lambda^{(0)}$, and $\rho$ are stemming from ALM and we follow the guideline from [2] when setting them. We initialize all the Lagrangian multipliers $\lambda_i^{(0)}$ to 0. We choose $\mu^{(0)}$ from the set $\{10^{-7}, 10^{-6}, 10^{-5}, 10^{-4}, 10^{-3}\}$, as it is suggested to choose a small value in the beginning and increase it iteratively using the equation $\mu^{(k+1)} = \rho \cdot \mu^{(k)}$. We

---

[4]Experiments with multiple critical classes are presented in supplementary materials.

choose $\rho$ from the set $\{2, 3\}$ as $\rho > 1$ is suggested. Moreover, we do not increase $\rho$ beyond 4 to avoid potential dominance of the constraint on $\bar{F}(\theta)$, since $\rho$ is used to increase $\mu$. Once we find the best combination of $\mu$ and $\rho$ based on the chosen metrics on the validation set, we fix them and we search for $\delta$ as final step. Please see supplementary materials for further details on the hyper-parameters selection.

Once the hyper-parameters for each model are selected, the training is performed and models are applied to the test set to yield the final results, which are described next.

Table 1: Results on binary CIFAR10 for class ratio 1:100 and 1:200.

| Dataset | Binary CIFAR10, imb. 100 | | | | Binary CIFAR10, imb. 200 | | | |
|---|---|---|---|---|---|---|---|---|
| Training method | FPR @ 98% TPR | FPR @ 95% TPR | FPR @ 92% TPR | Test AUC | FPR @ 98% TPR | FPR @ 95% TPR | FPR @ 92% TPR | Test AUC |
| BCE | 56.0 | 45.0 | 29.0 | 91.2 | 75.0 | 55.0 | 40.0 | **87.3** |
| S-ML | 59.0 | 40.0 | 26.0 | 91.7 | 75.0 | 54.0 | **35.0** | 87.4 |
| S-FL | 59.0 | 40.0 | 27.0 | **91.7** | 78.0 | 59.0 | 43.0 | 85.7 |
| A-ML | 54.0 | 36.0 | **23.0** | 92.4 | **74.0** | 56.0 | 39.0 | 87.4 |
| A-FL | 50.0 | 38.0 | 24.0 | 92.3 | **76.0** | 59.0 | 40.0 | 86.2 |
| CB-BCE | 89.0 | 72.0 | 59.0 | 78.0 | 87.0 | 74.0 | 61.0 | 78.0 |
| W-BCE | 69.0 | 52.0 | 37.0 | 87.4 | 88.0 | 75.0 | 62.0 | 78.3 |
| LDAM | 65.0 | 48.0 | 34.0 | 89.0 | 78.0 | 63.0 | 45.0 | **86.4** |
| MBAUC | 86.0 | 71.0 | 56.0 | 74.0 | 89.0 | 83.0 | 69.0 | 67.9 |
| ALM + BCE | **52.0** | **34.0** | **21.0** | **93.1** | **70.0** | 54.0 | 39.0 | 86.7 |
| ALM + S-ML | **50.0** | **37.0** | **24.0** | **92.5** | **72.0** | **52.0** | 39.0 | **87.9** |
| ALM + S-FL | **55.0** | **39.0** | **25.0** | 91.5 | **74.0** | **55.0** | 41.0 | **86.9** |
| ALM + A-ML | **45.0** | **35.0** | 23.0 | **92.8** | 75.0 | **74.0** | **35.0** | **87.6** |
| ALM + A-FL | **49.0** | **37.0** | **23.0** | **92.7** | 78.0 | **57.0** | **37.0** | **87.0** |
| ALM + CB-BCE | **67.0** | **51.0** | **36.0** | **88.1** | **85.0** | **69.0** | **53.0** | **80.0** |
| ALM + W-BCE | **66.0** | **48.0** | **31.0** | **89.3** | **83.0** | **69.0** | **54.0** | **81.0** |
| ALM + LDAM | **60.0** | **42.0** | **31.0** | **91.0** | **73.0** | **61.0** | **43.0** | 85.6 |

## 5.3 Results

We present binary classification results on CIFAR10, CIFAR100, and the in-house medical imaging datasets in Tables 1, 2 and 3, respectively. In the tables, we compare each baseline with the corresponding ALM+baseline and present the best result among the two by bold. Also, the underlined results indicate the best results among all. We evaluate the performance of the baselines and the proposed method (ALM) using FPR at maximal levels of TPR (or minimal levels of false negative rate (FNR)) and AUC on the test sets. In the binary experiments, ALM is overall able to consistently improve the performance of almost all the loss functions, with regard to both AUC and FPRs at maximal TPR levels. Even in those cases when AUC is improved to a moderate extent there is still an improvement in FPRs, in accordance with our goal. Moreover, it is noticeable that the higher the TPR, the higher the benefit of applying ALM, which is in accordance with our target applications. In fact, considering that such classifiers in "critical" applications would be operated at high TPR (or low FNR), reduction in FPR in these settings is the effect we desired from the proposed approach. Lastly, we also observe in the tables that directly optimizing AUC via MBAUC does not provide the same improvements as using the proposed ALM approach. We present additional binary classification experiments in the supplementary material.

Additionally, we present the quantitative results of multi-class experiments in Table 4. A bold result for a baseline means that it is able to outperform both constrained optimisation methods $ALM_{m,1}$ and $ALM_{m,2}$, otherwise the better constrained strategies are highlighted. As for the binary case, underlined results indicate the best method for each metric. Comparison with the other baselines are presented in the supplementary materials. In these experiments, we evaluate the performance by computing the accuracy on the non-critical classes, at various levels of TPR for the critical-class. For this purpose, the first step is to find a threshold on the logit of the critical class such that the desired

Table 2: Results on binary CIFAR100 for class ratio 1:100 and 1:200.

| Dataset | Binary CIFAR100, imb. 100 | | | | Binary CIFAR100, imb. 200 | | | |
|---|---|---|---|---|---|---|---|---|
| Training method | FPR @ 98% TPR | FPR @ 95% TPR | FPR @ 90% TPR | Test AUC | FPR @ 98% TPR | FPR @ 95% TPR | FPR @ 90% TPR | Test AUC |
| BCE | 93.0 | 63.0 | 47.0 | 81.8 | 94.0 | 77.0 | 61.0 | 79.1 |
| S-ML | 89.0 | **65.0** | 43.0 | **82.7** | 95.0 | 75.0 | 64.0 | 79.7 |
| S-FL | 89.0 | 62.0 | 44.0 | **82.6** | 90.0 | 78.0 | 50.0 | 80.1 |
| A-ML | 91.0 | 63.0 | 44.0 | 81.8 | 95.0 | 75.0 | 66.0 | 79.8 |
| A-FL | 88.0 | 63.0 | 45.0 | 82.8 | 91.0 | 78.0 | 50.0 | 80.0 |
| CB-BCE | 93.0 | 75.0 | 52.0 | 78.8 | 93.0 | 78.0 | 51.0 | 78.7 |
| W-BCE | 88.0 | 59.0 | 41.0 | 79.7 | 95.0 | 63.0 | 51.0 | 79.7 |
| LDAM | 84.0 | 70.0 | 42.0 | 82.8 | **80.0** | 67.0 | **45.0** | 82.1 |
| MBAUC | 81.0 | 62.0 | 41.0 | 82.3 | 88.0 | 63.0 | 48.0 | 80.3 |
| ALM + BCE | **91.0** | 49.0 | **39.0** | 82.7 | **87.0** | **66.0** | **57.0** | 80.9 |
| ALM + S-ML | **88.0** | 69.0 | **41.0** | 81.7 | **87.0** | **73.0** | **55.0** | 80.7 |
| ALM + S-FL | **88.0** | **60.0** | **42.0** | 81.7 | **85.0** | **76.0** | 50.0 | 80.8 |
| ALM + A-ML | **89.0** | **55.0** | **37.0** | 82.7 | **92.0** | **63.0** | **45.0** | 81.0 |
| ALM + A-FL | **86.0** | 62.0 | **40.0** | 83.2 | **88.0** | **76.0** | **46.0** | 80.7 |
| ALM + CB-BCE | **89.0** | **59.0** | 36.0 | 83.8 | **85.0** | **66.0** | 44.0 | 81.0 |
| ALM + W-BCE | **87.0** | **53.0** | **39.0** | 83.2 | 79.0 | **62.0** | 44.0 | 81.3 |
| ALM + LDAM | 80.0 | **59.0** | **40.0** | 83.2 | 84.0 | 61.0 | 46.0 | 81.5 |

TPR is obtained for the important class. All the test samples whose logit of critical class exceeds the selected threshold are assigned to the important class. The remaining samples that are not assigned to the critical class are then classified based on the highest probability over the non-critical logits. In addition to this metric, we present overall classification accuracy on all classes. The quantitative results demonstrate that both of the multi-class strategies, $ALM_{m,1}$ and $ALM_{m,2}$ improve on the baseline, reducing the error on non-critical classes, at high levels of accuracy on the important class. Moreover, $ALM_{m,2}$ is able to further improve the overall accuracy by a larger margin in almost all the experiments thanks to the additional term in the constraint.

Table 3: Results on in-house MRI dataset.

| Method | FPR @0 FN | FPR @1 FN | Avg AUC | AUC ens. |
|---|---|---|---|---|
| BCE | 80.0 | 80.0 | 65.4±9.0 | 70.9 |
| S-ML | 81.0 | 77.0 | 67.3±7.0 | 71.5 |
| S-FL | 77.0 | 38.0 | 71.7±10.0 | 80.3 |
| A-ML | 77.0 | 73.0 | 68.0±9.0 | 74.2 |
| A-FL | 66.0 | 38.0 | 67.7±8.0 | 80.1 |
| CB-BCE | 100.0 | **34.0** | 72.0±5.0 | 77.7 |
| W-BCE | **56.0** | **42.0** | 68.8±6.0 | 80.5 |
| LDAM | 100.0 | 75.0 | 62.0±9.0 | 66.4 |
| MBAUC | 61.2 | 33.0 | 71.2±11.0 | 82.4 |
| ALM + BCE | **54.0** | **38.0** | 76.8±9.0 | 85.4 |
| ALM + S-ML | **81.0** | **33.0** | 72.5±9.0 | **80.3** |
| ALM + S-FL | 53.0 | 26.0 | 72.5±10.0 | **84.2** |
| ALM + A-ML | **72.0** | **53.0** | 67.2±5.0 | **76.4** |
| ALM + A-FL | **62.0** | 46.0 | 74.7±7.0 | **81.5** |
| ALM + CB-BCE | **86.0** | 34.0 | 73.0±9.0 | **79.5** |
| ALM + W-BCE | 59.0 | **40.0** | 72.4±6.0 | **81.4** |
| ALM + LDAM | **59.0** | **53.0** | 66.5±8.5 | **77.0** |

Table 4: Results on long-tailed CIFAR10 for class imbalance 1:100 and 1:200. The Table shows the error on all the non-important classes, after setting a threshold on the logit of the important class to obtain 80, 90% TPR.

| Dataset | Long-tailed CIFAR10, imb. 100 | | | Long-tailed CIFAR10, imb. 200 | | |
|---|---|---|---|---|---|---|
| Training method | Error @ 80% TPR | Error @ 90% TPR | Overall Accuracy | Error @ 80% TPR | Error @ 90% TPR | Overall Accuracy |
| CE | 29.80 | 34.67 | 70.35 | 37.81 | 42.37 | 64.03 |
| FL | 32.21 | 36.47 | 69.20 | 38.50 | 42.21 | 62.90 |
| CB-BCE | 31.04 | 33.44 | **72.90** | 35.11 | 39.13 | 65.77 |
| LDAM | 26.57 | 29.86 | 71.80 | 34.41 | 45.72 | 65.87 |
| $ALM_{m,1}$ + CE | **28.89** | **33.93** | **70.90** | **36.14** | **39.90** | **65.13** |
| $ALM_{m,1}$ + FL | **29.59** | **34.94** | **69.74** | **36.87** | **41.87** | **64.23** |
| $ALM_{m,1}$ + CB-CE | **27.89** | **30.27** | 72.10 | **33.09** | 35.44 | 65.34 |
| $ALM_{m,1}$+ LDAM | **25.73** | **28.52** | **72.86** | **31.94** | **37.41** | 65.65 |
| $ALM_{m,2}$ + CE | 29.53 | 34.09 | 71.30 | **35.10** | **39.19** | 64.35 |
| $ALM_{m,2}$ + FL | 30.50 | 35.63 | 69.47 | **36.27** | **40.43** | 64.43 |
| $ALM_{m,2}$ + CB-CE | 27.84 | 31.97 | 72.09 | **33.72** | **36.86** | 66.04 |
| $ALM_{m,2}$ + LDAM | **24.76** | 28.91 | **73.32** | **31.02** | **36.09** | **67.41** |

# 6 Conclusion

In this paper, we pose the training of a DNN for binary classification under class imbalance as a constrained optimization problem and propose a novel constraint that can be used with existing loss functions. The proposed constraint is designed to maximize the AUC, but in an asymmetric way to favor the reduction of FPR at high TPR (or low FNR). Then, we transfer the constrained problem to its dual unconstrained optimization problem using an Augmented Lagrangian method (ALM) [2] which we optimize using stochastic gradient descent. Additionally, we present two possible extensions of the proposed constraint for multi-class classification problems.

We perform an extensive evaluation of the proposed constraints for binary and multi-class image classification problems on both computer vision and medical imaging datasets. We compare the performance of the proposed constraints with different baselines by simulating different class imbalance ratio. The quantitative results demonstrate that the proposed constraints improve the performance of the baselines in the majority of the cases in both binary and multi-class classification experiments.

# 7 Acknowledgments

The presented work was partly funding by: 1. Clinical Research Priority Program Grant on Artificial Intelligence in Oncological Imaging Network, University of Zurich and 2. Personalized Health and Related Technologies (PHRT), project number 222, ETH domain.

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
