# Constrained Optimization to Train Neural Networks on Critical and Under-Represented Classes

**Sara Sangalli**[1], **Ertunc Erdil**[1], **Andreas Hoetker**[2], **Olivio Donati**[2], **Ender Konukoglu**[1]
[1] Computer Vision Lab, ETH Zürich
[2] Institute for Diagnostic and Interventional Radiology, Universitätsspital Zürich
sara.sangalli@vision.ee.ethz.ch

## A    Additional Experiments

### A.1    Additional experiments on binary classification

In Table 1, we provide results for the additional class ratio on CIFAR10 of 1:50. The results demonstrate the effectiveness of the proposed method even at a lower class imbalance and show consistency with the results at higher imbalance presented in the main paper.

Table 1: Results on binary CIFAR10 for the same classes as in the main paper, at 1:50 class ratio.

| Dataset | Binary CIFAR10, imb. 50 | | | |
|---|---|---|---|---|
| Training method | FPR @ 98% TPR | FPR @ 95% TPR | FPR @ 90% TPR | Test AUC |
| BCE | 34.1 | 21.3 | 11.1 | 96.05 |
| S-ML | 31.6 | 18.0 | 9.2 | **96.48** |
| S-FL | 31.9 | 18.0 | **10.1** | **96.20** |
| A-ML | 28.4 | 17.0 | 10.6 | 96.29 |
| A-FL | 36.1 | 20.0 | 11.3 | 95.86 |
| CB-BCE | 84.6 | 72.3 | 55.4 | 79.73 |
| W-BCE | 36.9 | 22.5 | 12.8 | 95.29 |
| LDAM | 45.2 | 21.0 | 8.9 | 95.65 |
| MBAUC | 75.0 | 60.5 | 47.0 | 82.47 |
| ALM + BCE | **30.7** | **17.4** | **7.9** | **96.49** |
| ALM + S-ML | **31.1** | **17.0** | **8.9** | 96.41 |
| ALM + S-FL | **28.4** | **17.9** | 11.3 | 96.15 |
| ALM + A-ML | **28.7** | **16.1** | **9.2** | **96.40** |
| ALM + A-FL | **32.7** | **17.1** | **9.3** | **96.2** |
| ALM + CB-BCE | **52.9** | **35.5** | **25.8** | **91.78** |
| ALM + W-BCE | **32.4** | **18.4** | **11.2** | **95.87** |
| ALM + LDAM | **33.8** | **14.2** | **8.1** | **96.61** |

In Table 2, in addition to the experiments with 2 randomly selected classes of CIFAR10, we provide results for other two randomly selected classes. In this experiment, we present FPR results at higher TPRs compared to the results in the main paper because at lower thresholds, both baselines and ALM already perform quite well.

35th Conference on Neural Information Processing Systems (NeurIPS 2021).

Table 2: Results on binary CIFAR10 for class ratio 1:100 and 1:200 for two randomly selected classes different than the ones presented in the main paper.

| Dataset | Binary CIFAR10, imb. 100 | | | | Binary CIFAR10, imb. 200 | | | |
|---|---|---|---|---|---|---|---|---|
| Training method | FPR @ 100% TPR | FPR @ 99% TPR | FPR @ 96% TPR | Test AUC | FPR @ 100% TPR | FPR @ 99% TPR | FPR @ 96% TPR | Test AUC |
| BCE | 78.0 | **30.0** | 13.0 | 97.9 | 85.0 | 43.0 | 26.0 | 95.9 |
| S-ML | 78.0 | **34.0** | 14.0 | 97.9 | 98.0 | 62.0 | 25.0 | 95.2 |
| S-FL | 84.0 | 35.0 | **12.0** | 98.0 | 77.0 | **45.0** | 24.0 | **96.1** |
| A-ML | 91.0 | 34.0 | 14.0 | 97.9 | 85.0 | **52.0** | 27.0 | 95.7 |
| A-FL | 84.0 | 39.0 | 13.0 | 98.0 | 91.0 | 54.0 | 22.0 | 96.0 |
| CB-BCE | 81.0 | 59.0 | 36.0 | 92.6 | 81.0 | 57.0 | 35.0 | 92.2 |
| W-BCE | 87.0 | 55.0 | **24.0** | 95.0 | 97.0 | 68.0 | 44.0 | **91.6** |
| LDAM | 80.0 | 33.0 | 14.0 | 97.9 | 78.0 | 51.0 | **26.0** | 95.8 |
| MBAUC | 88.0 | 54.0 | 34.0 | 90.9 | 85.0 | 56.0 | 42.0 | 89.4 |
| ALM + BCE | **58.0** | 34.0 | **10.0** | 98.2 | **82.0** | **39.0** | **24.0** | 96.3 |
| ALM + S-ML | **72.0** | 34.0 | **12.0** | 98.0 | **92.0** | **52.0** | **24.0** | 96.1 |
| ALM + S-FL | **69.0** | **30.0** | 13.0 | 98.1 | **72.0** | 49.0 | 24.0 | 96.0 |
| ALM + A-ML | **78.0** | **25.0** | **11.0** | 98.2 | **83.0** | 54.0 | **26.0** | **96.0** |
| ALM + A-FL | **75.0** | **33.0** | 13.0 | 98.2 | **72.0** | **47.0** | **21.0** | 96.4 |
| ALM + CB-BCE | **76.0** | **49.0** | **26.0** | 95.3 | **70.0** | **47.0** | **27.0** | 95.1 |
| ALM + W-BCE | **82.0** | **54.0** | 25.0 | 95.3 | **86.0** | **61.0** | **42.0** | 91.0 |
| ALM + LDAM | **71.0** | **25.0** | **11.0** | 98.4 | **65.0** | **46.0** | 29.0 | 94.8 |

## A.2 Comparison with additional baselines for multi-class experiments on long-tailed CIFAR100

Due to space reason, in the main paper we report the comparison with only four methods for the multi-class experiments. In Table 3 we show the results on long-tailed CIFAR10 for other three baselines from the SoA. The results are consistent with the main paper where ALM improves the baselines in majority of the cases.

Table 3: Results on long-tailed CIFAR10 for class imbalance 1:100 and 1:200, comparing to other baselines. The table shows the error on all the non-important classes, after setting a threshold on the logit of the important class to obtain 80, 90% TPR as well as the overall accuracy.

| Dataset | Long-tailed CIFAR10, imb. 100 | | | Long-tailed CIFAR10, imb. 200 | | |
|---|---|---|---|---|---|---|
| Training method | Error @ 80% TPR | Error @ 90% TPR | Overall Accuracy | Error @ 80% TPR | Error @ 90% TPR | Overall Accuracy |
| S-LM | 30.69 | 35.09 | **71.94** | 38.17 | 41.74 | 64.49 |
| A-LM | 31.56 | 37.12 | 69.51 | 36.41 | 40.75 | 64.38 |
| A-FL | 29.80 | 65.33 | 70.35 | 36.12 | 41.26 | 64.12 |
| $ALM_{m,1}$ + S-LM | **29.90** | **33.70** | 71.61 | **36.37** | **38.96** | 64.20 |
| $ALM_{m,1}$ + A-LM | **30.07** | **35.49** | **70.32** | **35.23** | **39.18** | 64.14 |
| $ALM_{m,1}$ + A-FL | **28.97** | **34.13** | 70.20 | **35.04** | **38.63** | 64.71 |
| $ALM_{m,2}$ + S-LM | **29.62** | **33.92** | 70.85 | 37.93 | 40.74 | 64.81 |
| $ALM_{m,2}$ + A-LM | **29.24** | **34.73** | 71.69 | **34.96** | **38.65** | 65.08 |
| $ALM_{m,2}$ + A-FL | **28.70** | **32.73** | 71.27 | **34.59** | 39.70 | 64.52 |

## A.3 Multi-class experiments on long-tailed CIFAR100

Table 4 shows the results on long-tailed CIFAR100, for class ratios 1:20 and 1:50. We test on a lower imbalance ratio on CIFAR100, compared to CIFAR10, because there are only 500 samples per class in the original CIFAR100 and using higher ratios would mean having only a few critical class samples, which provides very little information.

Table 4: Results on long-tailed CIFAR100 for class imbalance 1:20 and 1:50. The table shows the error on all the non-important classes, after setting a threshold on the important class' logit to obtain 80, 90% TPR as well as the overall accuracy.

| Dataset | CIFAR100, imb. 20 | | | CIFAR10, imb. 50 | | |
|---|---|---|---|---|---|---|
| Training method | Error @ 80% TPR | Error @ 90% TPR | Overall Accuracy | Error @ 80% TPR | Error @ 90% TPR | Overall Accuracy |
| CE | 92.00 | 92.55 | 49.46 | 93.31 | 94.56 | 43.17 |
| FL | 91.96 | 92.69 | 49.73 | 92.33 | 93.94 | 42.96 |
| CB-BCE | 92.20 | 93.27 | 50.02 | 93.44 | 94.97 | 42.18 |
| LDAM | 92.01 | 92.57 | 49.60 | 93.83 | 94.93 | **44.29** |
| $ALM_{m,1}$ + CE | **91.39** | **91.87** | **50.82** | **92.60** | **93.50** | 43.45 |
| $ALM_{m,1}$ + FL | **91.40** | **91.95** | **50.27** | **92.25** | **92.99** | 43.52 |
| $ALM_{m,1}$ + CB-CE | **91.44** | **92.09** | **51.36** | **92.84** | **94.28** | 42.47 |
| $ALM_{m,1}$+ LDAM | **91.40** | **92.18** | **50.08** | **93.31** | **94.43** | 43.95 |
| $ALM_{m,2}$ + CE | **91.74** | **92.05** | **50.47** | **92.69** | **94.40** | 43.52 |
| $ALM_{m,2}$ + FL | **91.67** | **91.99** | **49.88** | **92.10** | **93.00** | 43.56 |
| $ALM_{m,2}$ + CB-CE | **91.75** | **92.60** | **50.73** | **92.66** | **93.48** | 42.43 |
| $ALM_{m,2}$ + LDAM | **91.65** | **92.38** | **50.10** | **92.61** | **93.22** | 43.74 |

## A.4 Experiments with multiple critical classes

So far, we have performed experiments for the cases where there is only a single critical class. However, in practice, there may be multiple critical classes where missing a sample has a high cost. In Table 5 we present results when there are two critical, under-represented classes. In this case, we report the accuracy of each of the two important classes along with the accuracy on the non-critical ones. Overall, the proposed method is able to improve in almost all the cases the accuracy on the important classes, keeping a comparable accuracy in all the non-critical ones.

## A.5 Additional categories of Related Work

In the main paper we present related work about learning with class imbalance belonging to the cost sensitive training-based methods category. We decided to focus on this first because the presented method belongs to this category, and consequently it has been the focus of the discussion, secondly because of space reasons. In this Section we present the other two main categories of techniques that aim to address this aspect: sampling-based and classifier-based methods.

**Sampling-based methods:** Methods in this group aim to deal with the data imbalance problem by generating a balanced distribution through getting more samples from the minority class or less samples from the majority class. A simple approach of replicating a certain number of instances from the minority class can lead to models that are over-fitting to the over-sampled instances. [6] proposes to generate novel samples from minority class by interpolating the neighboring data points. [9] extends [6] by proposing a way to estimate the number of samples of the minority class to be synthesized. [8] approaches the problem from the opposite perspective and randomly under-sample majority class instances instead of synthesizing new data for the minority class. Despite the fact that losing valuable information for the majority class, [8] reports that it leads to better results compared to the former approaches. Although, these earlier sampling-based methods are useful for the low dimensional data, they suffer from issues in higher dimensions, e.g. images, since interpolation does not lead to realistic samples. Moreover, they still suffer from generalization difficulties [10]. [16]

Table 5: Results on long-tailed CIFAR10 for class imbalance 1:100 and 1:200, when the two smallest classes are identified as critical and under-represented. The table shows the accuracy on the two critical classes, the smallest identified as Class 1 and the other named Class 2, as well the accuracy over all the non-critical ones.

| Dataset | CIFAR10, imb. 100 | | | CIFAR10, imb. 200 | | |
|---|---|---|---|---|---|---|
| Training method | Acc. Critical Class 1 | Acc. Critical Class 2 | Acc. Other Classes | Acc. Critical Class 1 | Acc. Critical Class 2 | Acc. Other Classes |
| CE | 38.9 | 48.0 | 77.1 | 23.1 | 32.0 | 73.0 |
| FL | 37.5 | 37.0 | 76.2 | 22.7 | 24.5 | 72.7 |
| CB-BCE | 50.6 | **58.6** | 77.0 | 30.6 | 37.3 | 73.7 |
| LDAM | 48.9 | 44.5 | **78.1** | 33.1 | 30.6 | 74.4 |
| $\text{ALM}_{m,1}$ + CE | **43.0** | **50.0** | 77.0 | **32.0** | 29.0 | **73.7** |
| $\text{ALM}_{m,1}$ + FL | **41.0** | **41.6** | **76.6** | **25.8** | **31.5** | 72.6 |
| $\text{ALM}_{m,1}$ + CB-CE | **53.9** | 51.2 | **77.2** | **38.3** | **39.7** | **74.0** |
| $\text{ALM}_{m,1}$+ LDAM | **50.3** | **44.5** | 78.0 | **35.2** | **31.5** | 74.4 |
| $\text{ALM}_{m,2}$ + CE | **41.1** | **54.1** | 77.3 | **30.1** | **32.1** | 73.5 |
| $\text{ALM}_{m,2}$ + FL | **38.9** | **48.3** | 76.8 | **28.1** | **26.6** | 72.8 |
| $\text{ALM}_{m,2}$ + CB-CE | **53.6** | 56.7 | 76.6 | **33.8** | **45.3** | 73.1 |
| $\text{ALM}_{m,2}$ + LDAM | **50.0** | **47.1** | 78.0 | **34.5** | 30.4 | **74.6** |

proposes to use adversarial training with capsule networks to generate more realistic samples for the minority classes, and demonstrate its effectiveness for class imbalance. More recently, [14] proposes a method which adaptively samples a subset from the training set in each iteration to train multiple classifiers which are then ensembled for prediction.

**Classifier-based methods:** Methods from this category operate in test time and are mostly based on thresholding and scaling the output class probabilities. One common approach is to divide the output for each class by their prior probabilities which shown to be effective to handle class imbalance in both classification [13, 3] and semantic segmentation [5]. In a recent work, [17] argue that the previous methods from this family suffer from diminished overall accuracy despite the improved detection on minority classes. They mitigate this problem by proposing a method re-balancing the posterior in test-time.

## B Background = Augmented Lagrangian Method

The Augmented Lagrangian Method is based on two previously developed techniques, which are combined together into ALM, overcoming the respective drawbacks.

A generic optimization problem for an objective function $F(\theta)$ subject to the constraints $\mathcal{C}(\theta) = \{c_1(\theta), ..., c_m(\theta)\}$ can be expressed as [2, 15]:

$$\begin{aligned} \arg\min_{\theta} F(\theta) \\ \text{subject to } \mathcal{C}(\theta), \quad \theta \in \Theta \end{aligned} \tag{1}$$

One of the earlier methods, quadratic penalty method [1], converts the constrained optimization problem in Eq. (1) to an unconstrained optimization problem by adding the constraint to the objective function as a quadratic penalty term:

$$\arg\min_{\theta \in \Theta} F(\theta) + \mu \sum_{i=1}^{m} \|c_i(\theta)\|^2 \tag{2}$$

where $\mu$ is a positive parameter which controls the contribution of the penalty term to the overall loss function. Increasing $\mu$ indefinitely over the iterations is necessary to convexify the loss and ensure convergence. However, as $\mu$ increases, the penalty term prevails $F(\theta)$, which makes training unstable [1].

The method of Lagrange multipliers converts Eq. (1) into the unconstrained optimization problem by adding the constraints to the objective function as follows:

$$\mathcal{L}(\theta, \lambda) = F(\theta) + \sum_{i=1}^{m} \lambda_i c_i(\theta) \tag{3}$$

where $\lambda$ are called as Lagrange multipliers. The method of Lagrange multipliers deals with the instability of quadratic penalty method, however, it requires the objective function to be convex which is a drawback.

Augmented Lagrangian Methods overcome the limitations of the above-mentioned two approaches. Here, the penalty concept is merged with the primal-dual philosophy of classic Lagrangian function, as explained in the main paper.

## C Theoretical Insights

In this Section we derive a few theoretical insights to motivate the choice of our constraint with the aid of an example. Let us consider the situation depicted in Figure 1 of a binary classification task, where the NN is correctly ordering all the samples, except for one mistake for each class. This case can be then generalised for a larger number of mistakes. For simplicity, let us consider $\Delta$ as the distance between all adjacent pairs of correct samples, $\Delta_N$ as distance between the misclassified negative sample and the rightmost positive sample and finally $\Delta_P$ as the distance between the misclassified positive sample and the leftmost negative sample. Let M and N be the number of positive and negative examples, respectively and $M \leq N$, i.e. positive class is smaller (or equal) than negative class, which corresponds to the setup of interest. Our goal is to show that: 1) the proposed constraint encourages the loss to reduce the error on the positive sample (i.e. improving TPR) instead of the negative one; 2) the other asymmetric variant (i.e. $\sum_{j=1}^{M} \max(0, -(f_\theta(x_j^+) - f_\theta(x_k^-)) + \delta) = 0, \; k \in \{1, \ldots, N\}$) wouldn't have been as effective as the chosen one.

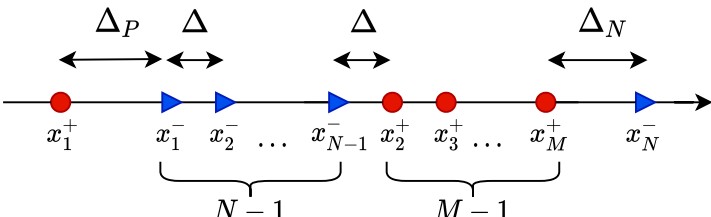

Figure 1: Auxiliary example to explain the motivations behind the chosen constraint. All the correctly ranked samples are separated by a distance $\Delta$, while the errors amount to $\Delta_N$ and $\Delta_P$, for the negative and positive errors respectively.

Accordingly to our solution the update of multiplier for each positive training sample is:

$$\lambda_1 = \lambda + \mu \left( \sum_{k=0}^{N-2} \Delta_P + k\Delta \right)$$

$$\lambda_{M-j} = \lambda + \mu(\Delta_N + \Delta j) \quad j = 0, ..., M - 2 \tag{4}$$

Consequently, the loss terms corresponding to ALM are:

$$
\tilde{L}_\mu(\theta, \lambda) = \frac{\mu}{2} \left[ \left( \sum_{k=0}^{N-2} \Delta_P + k\Delta \right)^2 + \sum_{j=0}^{M-2} (\Delta_N + j\Delta)^2 \right]
$$

$$
+ \left[ \left( \lambda + \mu \sum_{k=0}^{N-2} \Delta_P + k\Delta \right) \left( \sum_{k=0}^{N-2} \Delta_P + k\Delta \right) + \sum_{j=0}^{M-2} (\lambda + \mu(\Delta_N + j\Delta))(\Delta_N + j\Delta) \right] =
$$

$$
= \frac{3\mu}{2} \left( \sum_{k=0}^{N-2} \Delta_P + k\Delta \right)^2 + \lambda \sum_{k=0}^{N-2} (\Delta_P + k\Delta) + \frac{3\mu}{2} \sum_{j=0}^{M-2} (\Delta_N + j\Delta)^2 + \lambda \sum_{j=0}^{M-2} (\Delta_N + j\Delta) =
$$

$$
= \frac{3\lambda}{2} \left[ \Delta_P(N-1) + \Delta \frac{(N-1)(N-2)}{2} \right]^2 + \Delta_P \lambda (N-1) + \lambda \Delta \frac{(N-1)(N-2)}{2}
$$

$$
+ \frac{3\mu}{2} \left[ \Delta_N^2 (M-1) + \Delta^2 \frac{(2M-3)(M-1)(M-2)}{6} + \Delta_N 2\Delta \frac{(M-1)(M-2)}{2} \right]
$$

$$
+ \Delta_N \lambda (M-1) + \lambda \Delta \frac{(M-1)(M-2)}{2} =
$$

$$
= \frac{3\mu}{2} \left[ \Delta_P^2 (N-1)^2 + \Delta_N^2 (M-1) \right]
$$

$$
+ \Delta_P \left[ \frac{3}{2} \mu \Delta (N-1)^2 (N-2) + \lambda(N-1) \right] + \Delta_N \left[ \frac{3}{2} \mu \Delta (M-1)(M-2) + \lambda(M-1) \right]
$$

$$
+ \frac{3}{2} \mu \Delta^2 \left( \frac{((N-1)(N-2))^2}{4} + \frac{(2M-3)(M-1)(M-2)}{6} \right)
$$

$$
+ \lambda \Delta \left( \frac{(N-1)(N-2)}{2} + \frac{(M-1)(M-2)}{2} \right)
$$

(5)

**Insight 1:** It is evident from the final result of Equation 5 that, given $M \leq N$ (in our experiments the inequality is strict) and given the same error ($\Delta_P = \Delta_N$), the contribution to the loss function from the misclassified positive sample is larger than the contribution of the negative one. This means that we are putting more emphasis on errors in the smaller class, even when the entity of the mistake is the same for both classes. As a consequence, removing the error $\Delta_P$ would reduce the loss more than removing the error $\Delta_N$. Moreover, it is clear that this difference in error weighing increases with the level of imbalance between the classes. This results not only from the different number of samples per class, but also from the presence of higher powers in the coefficients of $\Delta_P$ in Equation 5, i.e. $(N-1)^2$ in both the terms with $\Delta_P$. Clearly, this consideration holds even more in the case where $\Delta_P \geq \Delta_N$, as the difference is further enforced.

Differently, if $\Delta_N \geq \Delta_P$ it is not always guaranteed that removing the error $\Delta_P$ reduces the loss more than removing the error $\Delta_N$. Fixing $\Delta_P$, M, and N (with $M \leq N$), $\Delta_N$ may be increased such that its contribution is higher than the one from the positive sample. In order to visually understand this, Figure 2 shows two possible contributions to the loss from the positive ($L_P$) and negative ($L_N$) errors. At the same $\Delta_P = \Delta_N = \Delta_E$ or at $\Delta_P \geq \Delta_N$ the contribution of the mistake on the positive example is higher. However, this does not hold anymore when, fixed $\Delta_P$, $\Delta_N$ exceeds a certain $\Delta_P + \Delta_{diff,lim}$. In this situation, when $\Delta_N \geq \Delta_P + \Delta_{diff,lim}$, the condition $L_P \geq L_N$ does not hold anymore.

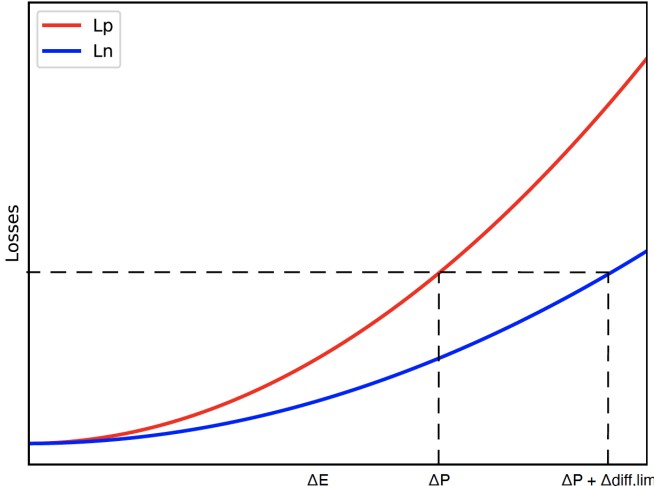

Figure 2: Trend of the two contributions of positive and negative errors.

In order to find $\Delta_{diff,lim}$ it is necessary that the following equation for $\Delta_{diff}$ is satisfied:

$$
\begin{aligned}
\Delta_P^2(N-1)^2 + \Delta_P\left[\frac{3}{2}\mu\Delta(N-1)^2(N-2) + \lambda(N-1)\right] = \\
(\Delta_P + \Delta_{diff})^2(M-1) + (\Delta_P + \Delta_{diff})\left[\frac{3}{2}\mu\Delta(M-1)(M-2) + \lambda(M-1)\right]
\end{aligned}
\tag{6}
$$

For readability Equation 6 may be rewritten as:

$$
\Delta_P^2 a + \Delta_P b = (\Delta_P + \Delta_{diff})^2 c + (\Delta_P + \Delta_{diff})d
\tag{7}
$$

Which leads to:

$$
\begin{aligned}
c\Delta_{diff}^2 + (2\Delta_P c + d)\Delta_{diff} + (\Delta_P^2(c-a) + \Delta_P(d-b)) = 0 \\
\Delta_{diff,lim} = \frac{-(2\Delta_P c + d) + \sqrt{(2\Delta_P c + d)^2 - 4c(\Delta_P^2(c-a) + \Delta_P(d-b))}}{2c}
\end{aligned}
\tag{8}
$$

We take only the positive square root as $\Delta_{diff,lim}$ has to be positive and the discriminant of the equation is positive for $N \geq M$. A few considerations can be drawn about how $\Delta_{diff,lim}$ varies. The higher the class imbalance, the higher $\Delta_{diff,lim}$ becomes (as this affects the discriminant in the equation). Similarly, the larger the fixed $\Delta_P$, the larger $\Delta_{diff,lim}$.

However in practice, being NNs typically biased towards majority class, it is likely that this last case of $\Delta_P < \Delta_N$ rarely occurs.

**Insight 2:** Let us now show that choosing the symmetrically opposite constraint would have been less efficient than the proposed method for our purpose. Accordingly to the alternative version of the constraint, the Lagrange multipliers would have been updated as follows:

$$
\begin{aligned}
\lambda_{k+1} = \lambda + \mu(\Delta_N + \Delta k) \quad k = 0, ..., N-2 \\
\lambda_N = \lambda + \mu\left(\sum_{j=0}^{M-2} \Delta_N + j\Delta\right)
\end{aligned}
\tag{9}
$$

And the loss function would have become:

$$
\tilde{L}_\mu(\theta, \lambda) = \frac{\mu}{2} \left[ \left( \sum_{j=0}^{M-2} \Delta_N + j\Delta \right)^2 + \sum_{k=0}^{N-2} (\Delta_P + j\Delta)^2 \right]
$$

$$
+ \left[ \left( \lambda + \mu \sum_{j=0}^{M-2} \Delta_N + j\Delta \right) \left( \sum_{j=0}^{M-2} \Delta_N + j\Delta \right) + \sum_{k=0}^{N-2} (\lambda + \mu(\Delta_P + k\Delta))(\Delta_P + k\Delta) \right] =
$$

$$
= \frac{3\mu}{2} \left( \sum_{j=0}^{M-2} \Delta_N + j\Delta \right)^2 + \lambda \sum_{k=0}^{N-2} (\Delta_P + k\Delta) + \frac{3\mu}{2} \sum_{k=0}^{N-2} (\Delta_P + k\Delta)^2 + \lambda \sum_{j=0}^{M-2} (\Delta_N + j\Delta) =
$$

$$
= ... =
$$

$$
= \frac{3\mu}{2} \left[ \Delta_P^2 (N-1) + \Delta_N^2 (M-1)^2 \right]
$$

$$
+ \Delta_P \left[ \frac{3}{2} \mu \Delta (N-1)(N-2) + \lambda(N-1) \right] + \Delta_N \left[ \frac{3}{2} \mu \Delta (M-1)^2 (M-2) + \lambda(M-1) \right]
$$

$$
+ \frac{3}{2} \mu \Delta^2 \left( \frac{((M-1)(M-2))^2}{4} + \frac{(2N-3)(N-1)(N-2)}{6} \right)
$$

$$
+ \lambda \Delta \left( \frac{(N-1)(N-2)}{2} + \frac{(M-1)(M-2)}{2} \right)
$$

$$(10)$$

The difference between the proposed method and this alternative resides in the coefficients that multiply the terms in $\Delta_P$ and $\Delta_N$. When $M \leq N$, looking at Equation 5, it emerges that our loss weighs more the positive error than the alternative constraint in Equation 10, thanks to the coefficients of $\Delta_P$, which are smaller for the alternative loss.

## D    Additional training details

### D.1    Network details

For the binary classification on the 3D MRI images, we employ a DNN architecture composed by two, identical and parallel structures. For each patient, one path of the NN processes the diffusion-weighted images and the other part processes the T2-weighted axial images. Each path consists of cascaded 3D convolution, 3D max-pooling, and activation functions. Finally, in the fully connected layer the outputs are concatenated, as the different modalities carry complementary information. In the binary classification on CIFAR10 and CIFAR100, we use ResNet-10 [11] trained for 100 epochs and Adam optimizer, without any learning rate schedule and a batch size of 64. The common hyperparameters are selected based on the best validation AUC of BCE. For the multi-class, long-tailed experiments we follow the setup used by [4], as it is a recent and acknowledged work from the state of the art, and we keep it consistent over the baselines and ALM experiments, for a fair comparison. Specifically, we use ResNet-32 [11] as our base network, and use stochastic gradient descend with momentum of 0.9, weight decay of $2 \times 10^{-4}$ for training. The model is trained with a batch size of 128 for 200 epochs. We use an initial learning rate of 0.1, then decay by 0.01 at the 160th epoch and again at the 180th epoch.

### D.2    Further details on hyperparameters

We search the hyperparameter space incrementally in order to slightly reduce the number of simulations we run for computational purposes. Common hyperparameters such as number of epochs, learning rate, batch size and patience for early stopping have been selected as explained in Section D. Then, these parameters are kept fixed for all the experiments. Next, for the baselines having specific hyperparameters, the search is carried out specifically for each dataset as well as for each class ratio, among those values proposed by the original papers (apart from W-BCE for which does not provide

specific guideline for setting the weight of the loss, thus we decided to set it proportionate to the class ratio $N$ for samples of minority class). More specifically, for the baselines the following values are considered:

- margin $m$ has been searched among {0.5, 2, 4} for S-ML and A-ML

- exponent $\gamma$ has been searched among {0.5, 1, 2} for S-FL and A-FL

- weight coefficient $m$ has been searched among {N/3, 2N/3, N } for WBCE, being $N$ the class ratio

- exponent $\beta$ has been searched among {0.99, 0.999, 0.9999} for cb-BCE

Once the hyperparameters of the baselines are set, we keep them fixed for ALM training. Then, we seek for for the best hyperparameters for ALM. With the same logic, we perform a grid search with varying $\rho$ and $\mu^{(0)}$. We choose $\mu^{(0)}$ from the set $\{10^{-7}, 10^{-6}, 10^{-5}, 10^{-4}, 10^{-3}\}$ and $\rho$ from the set $\{2, 3\}$, but we found that selecting the smallest $\rho$ always provided the best results. Once we find the best combination of $\mu$ and $\rho$ based on the AUC on the validation set, we fix them and search for $\delta$ from the set {0.1, 0.25, 0.5, 1.0} for the binary task and {0.05, 0.1} for the multi-class set-up.

# E  Ensembling results for binary CIFAR10 and CIFAR100

In Tables 6 and 7 we report the test AUC obtained with ensembling (the same as Tables 1 and 2 of the main paper), along with the corresponding average and the standard deviation of AUC over the 10 runs, for both the binary CIFAR10 and CIFAR100 experiments. Comparing results in Tables 6 and 7 with those for the MRI dataset in Table 3 of the main paper, it is noticeable that the standard deviation is consistently larger for the MRI dataset, compared to CIFAR10 and CIFAR100, reflecting a higher uncertainty of the network on the predictions for this dataset [12].

Table 6: Avg AUC over 10 runs and the corresponding ensembled AUC (reported in the main paper) for CIFAR10 in binary classification.

| Dataset | CIFAR10, imb. 100 | | CIFAR10, imb. 200 | |
|---|---|---|---|---|
| Training method | Avg. AUC | Ens. AUC | Avg. AUC | Ens. AUC |
| BCE | $82.0 \pm 2.6$ | 91.2 | $75.8 \pm 4.0$ | 87.3 |
| S-ML | $82.6 \pm 2.4$ | 91.7 | $75.6 \pm 3.5$ | 87.4 |
| S-FL | $82.7 \pm 2.0$ | 91.7 | $74.8 \pm 3.4$ | 85.7 |
| A-ML | $83.4 \pm 2.3$ | 92.4 | $76.2 \pm 2.5$ | 87.4 |
| A-FL | $82.9 \pm 2.3$ | 92.3 | $74.6 \pm 4.1$ | 86.2 |
| CB-BCE | $73.2 \pm 2.0$ | 78.3 | $70.2 \pm 3.6$ | 78.1 |
| W-BCE | $79.0 \pm 2.9$ | 87.4 | $68.7 \pm 3.3$ | 78.3 |
| LDAM | $78.1 \pm 3.3$ | 89.0 | $74.1 \pm 3.4$ | 86.4 |
| MBAUC | $70.2 \pm 4.2$ | 74.0 | $63.7 \pm 3.9$ | 67.9 |
| ALM + BCE | $83.6 \pm 1.6$ | 93.1 | $75.5 \pm 3.3$ | 86.7 |
| ALM + S-ML | $83.6 \pm 1.7$ | 92.5 | $76.4 \pm 3.5$ | 87.9 |
| ALM + S-FL | $82.2 \pm 1.6$ | 91.5 | $75.8 \pm 3.3$ | 86.9 |
| ALM + A-ML | $83.6 \pm 2.1$ | 92.8 | $76.5 \pm 3.4$ | 87.6 |
| ALM + A-FL | $82.7 \pm 2.3$ | 92.7 | $75.6 \pm 3.7$ | 87.0 |
| ALM + CB-BCE | $76.2 \pm 5.3$ | 88.1 | $71.6 \pm 3.2$ | 80.6 |
| ALM + W-BCE | $80.6 \pm 1.9$ | 89.3 | $72.2 \pm 3.0$ | 81.0 |
| ALM + LDAM | $80.0 \pm 3.1$ | 91.0 | $74.2 \pm 2.7$ | 85.6 |

Table 7: Avg AUC over 10 runs and the corresponding ensembled AUC (reported in the main paper) for CIFAR100 in binary classification.

| Dataset | CIFAR100, imb. 100 | | CIFAR100, imb. 200 | |
|---|---|---|---|---|
| Training method | Avg. AUC | Ens. AUC | Avg. AUC | Ens. AUC |
| BCE | $77.8 \pm 1.7$ | 81.8 | $75.7 \pm 3.6$ | 79.1 |
| S-ML | $78.0 \pm 2.5$ | 82.7 | $75.4 \pm 4.0$ | 79.7 |
| S-FL | $78.2 \pm 2.7$ | 82.6 | $75.7 \pm 3.6$ | 80.1 |
| A-ML | $77.8 \pm 1.7$ | 81.8 | $75.8 \pm 3.8$ | 79.8 |
| A-FL | $78.1 \pm 2.3$ | 82.8 | $75.7 \pm 3.4$ | 80.1 |
| CB-BCE | $76.7 \pm 1.6$ | 78.8 | $74.9 \pm 3.3$ | 78.7 |
| W-BCE | $76.6 \pm 1.6$ | 79.7 | $76.4 \pm 1.7$ | 79.7 |
| LDAM | $77.6 \pm 3.8$ | 82.8 | $76.9 \pm 2.4$ | 82.1 |
| MBAUC | $79.7 \pm 1.5$ | 82.3 | $78.6 \pm 1.6$ | 80.3 |
| ALM + BCE | $78.6 \pm 1.8$ | 82.7 | $74.7 \pm 4.3$ | 80.9 |
| ALM + S-ML | $78.8 \pm 1.8$ | 81.7 | $74.9 \pm 3.7$ | 80.7 |
| ALM + S-FL | $78.0 \pm 2.7$ | 81.7 | $74.7 \pm 3.8$ | 80.8 |
| ALM + A-ML | $78.2 \pm 2.4$ | 82.7 | $76.4 \pm 3.2$ | 81.0 |
| ALM + A-FL | $77.9 \pm 2.7$ | 83.2 | $75.8 \pm 2.9$ | 80.8 |
| ALM + CB-BCE | $79.5 \pm 1.6$ | 83.8 | $77.7 \pm 2.2$ | 81.0 |
| ALM + W-BCE | $79.8 \pm 0.8$ | 83.2 | $76.5 \pm 2.6$ | 81.3 |
| ALM + LDAM | $77.8 \pm 2.6$ | 83.2 | $78.4 \pm 2.1$ | 81.5 |

# F   Statistical significance

All the results presented in the paper are obtained by averaging 10 runs with different random seeds for the model parameters. We perform statistical significance analysis on the AUC results using the DeLong test [7]. We copy the results of ALM below from Tables 1, 2, and 3 in the main paper and marked the ones that passes the DeLong test ($p \leq 0.5$) using *. Also, note that we wrote the results where ALM improves baseline using bold font. Therefore, a bold result marked with * indicates that the improvement achieved by ALM over the baseline is statistically significant which is the case in the majority of the cases shown in the Tables 8, 9 and 10.

Table 8: Statistical significance analysis on binary CIFAR10 for class ratio 1:100 and 1:200.

| Dataset | Binary CIFAR10, imb. 100 | Binary CIFAR10, imb. 200 |
|---|---|---|
| Training method | AUC | AUC |
| ALM + BCE | **93.1*** | 86.7 |
| ALM + S-ML | **92.5*** | **87.9*** |
| ALM + S-FL | 91.5 | **86.9*** |
| ALM + A-ML | **92.8** | **87.6*** |
| ALM + A-FL | **92.7*** | **87.0*** |
| ALM + CB-BCE | **88.1*** | **80.0*** |
| ALM + W-BCE | **89.3*** | **81.0*** |
| ALM + LDAM | **91.0*** | 85.6 |

Table 9: Statistical significance analysis on binary CIFAR100 for class ratio 1:100 and 1:200.

| Dataset | Binary CIFAR100, imb. 100 | Binary CIFAR100, imb. 200 |
|---|---|---|
| Training method | AUC | AUC |
| ALM + BCE | **82.7*** | **80.9*** |
| ALM + S-ML | 81.7 | **80.7*** |
| ALM + S-FL | 81.7 | **80.8*** |
| ALM + A-ML | **82.7*** | **81.0*** |
| ALM + A-FL | **83.2*** | **80.7** |
| ALM + CB-BCE | **83.8*** | **81.0*** |
| ALM + W-BCE | **83.2** | **81.3*** |
| ALM + LDAM | **83.2** | 81.5 |

Table 10: Statistical significance analysis on in-house MRI dataset.

| Method | AUC ens. |
|---|---|
| ALM + BCE | **85.4*** |
| ALM + S-ML | **80.3*** |
| ALM + S-FL | **84.2*** |
| ALM + A-ML | **76.4*** |
| ALM + A-FL | **81.5** |
| ALM + CB-BCE | **79.5** |
| ALM + W-BCE | **81.4*** |
| ALM + LDAM | **77.0*** |

# G   Results on MRI dataset at higher levels of False Negatives

In this Section we report the FPRs on the MRI dataset @2FNs, and @5FNs. Overall, it can be observed that the largest benefit from ALM is obtained at higher TPR, consistently with our goal.

Table 11: Results on in-house MRI dataset at higher FNRs.

| Method | FPR @2 FNs | FPR @5 FNs |
|---|---|---|
| BCE | 31.3 | 12.5 |
| S-ML | 37.5 | 9.4 |
| S-FL | 25.0 | 4.5 |
| A-ML | 26.6 | 4.5 |
| A-FL | 21.8 | 4.5 |
| CB-BCE | 23.4 | 1.6 |
| W-BCE | 32.8 | 4.5 |
| LDAM | 25.0 | 4.5 |
| ALM + BCE | **20.3** | **1.6** |
| ALM + S-ML | **21.8** | **1.6** |
| ALM + S-FL | **21.5** | **1.6** |
| ALM + A-ML | **21.8** | 4.5 |
| ALM + A-FL | **15.6** | **1.6** |
| ALM + CB-BCE | **21.5** | 1.6 |
| ALM + W-BCE | **21.8** | **1.6** |
| ALM + LDAM | **21.5** | **1.6** |

# H    Additional experiments at lower and consistent imbalances

In an earlier version of this work, we tested ALM on a smaller version of CIFAR10 and injected class imbalance with ratios 1:2, 1:9, and 1:19. Moreover, the previous setting presented a consistent imbalance among training, validation and test sets. We provide the results obtained from the previous study in the Table 12.

Table 12: Evaluation of results on CIFAR10 dataset. From top to bottom, results are shown for class ratio 1:2, 1:9, 1:19 with consistent imbalance across training, validation and test sets.

| Training method | Test AUC | FPR @ 100% TPR | FPR @ 95% TPR | FPR @ 90% TPR |
|---|---|---|---|---|
| BCE | 94.67 | 69.1 | **20.9** | 15.2 |
| S-ML | 94.57 | 68.2 | **21.6** | 15.2 |
| S-FL | 94.74 | 64.3 | 22.0 | 15.6 |
| A-ML | 94.56 | 67.0 | 21.8 | 15.6 |
| A-FL | 94.87 | 65.1 | **20.1** | **13.8** |
| W-BCE | 94.54 | 68.7 | 21.9 | 15.6 |
| CB-BCE | 94.38 | 69.3 | 23.5 | 16.3 |
| MBAUC | 94.26 | 69.6 | 23.80 | 15.40 |
| ALM + BCE | **95.41** | **66.2** | 21.1 | **13.2** |
| ALM + S-ML | **95.10** | **61.9** | 21.9 | **13.5** |
| ALM + S-FL | **95.22** | **54.3** | **20.5** | 14.7 |
| ALM + A-ML | **95.18** | **65.0** | 21.4 | **14.3** |
| ALM + A-FL | **94.95** | **64.0** | 20.5 | 14.7 |
| ALM + W-BCE | **95.67** | **59.9** | **18.7** | **13.2** |
| ALM + CB-BCE | **95.47** | **58.8** | **18.5** | **13.4** |
| BCE | 93.96 | 41.3 | 20.6 | 16.6 |
| S-ML | 94.04 | 39.6 | 20.3 | 16.2 |
| S-FL | 93.39 | 39.7 | **19.4** | 17.6 |
| A-ML | 93.64 | 42.1 | 21.3 | 17.3 |
| A-FL | 93.70 | 42.0 | 20.9 | 17.1 |
| W-BCE | 91.12 | 54.1 | 30.5 | 23.4 |
| CB-BCE | 90.83 | 58.5 | 31.7 | 27.1 |
| MBAUC | 92.04 | 44.1 | 22.7 | 17.0 |
| ALM + BCE | **94.74** | **34.2** | **19.9** | **14.1** |
| ALM + S-ML | **94.98** | **28.5** | **20.0** | **14.0** |
| ALM + S-FL | **94.2** | **35.6** | 22.5 | **17.4** |
| ALM + A-ML | **94.87** | **32.9** | **18.9** | **13.6** |
| ALM + A-FL | **95.38** | **31.4** | **16.5** | **12.4** |
| ALM + W-BCE | **93.03** | **47.9** | **23.1** | **19.8** |
| ALM + CB-BCE | **93.89** | **38.5** | **22.7** | **17.9** |
| BCE | 91.95 | 40.4 | 27.9 | 21.1 |
| S-ML | 92.28 | 36.4 | 27.9 | 22.0 |
| S-FL | 92.17 | 39.3 | 23.5 | 22.7 |
| A-ML | 91.74 | 34.2 | 27.4 | 22.6 |
| A-FL | 91.88 | 45.8 | 32.5 | **21.7** |
| W-BCE | 88.85 | 61.2 | 41.2 | 32.7 |
| CB-BCE | 88.24 | 61.6 | **36.5** | 35.1 |
| MBAUC | 91.8 | 36.00 | 26.4 | 25.0 |
| ALM + BCE | **93.21** | **29.8** | **27.5** | **17.2** |
| ALM + S-ML | **93.76** | **28.4** | **24.1** | **17.9** |
| ALM + S-FL | **93.50** | **31.0** | 23.5 | **16.8** |
| ALM + A-ML | **93.06** | **29.0** | **26.2** | **22.4** |
| ALM + A-FL | **93.45** | **34.4** | **27.7** | 22.3 |
| ALM + W-BCE | **91.22** | **48.2** | **31.2** | **19.6** |
| ALM + CB-BCE | **90.80** | **52.6** | 42.1 | **25.7** |