# OpenReview forum: "Constrained Optimization to Train Neural Networks on Critical and Under-Represented Classes"
_NeurIPS.cc/2021/Conference — NeurIPS 2021 Poster_

### Official Review · Reviewer_47qX · 2021-07-13

**Rating:** 7
**Confidence:** 3

**Summary:**

The paper proposes a novel method to perform classification with high imbalanced classes. The aim of the method is to minimise the FPR at high TPR -- this is suitable for many safety-critical applications where a misclassification can have dire consequences. In order to solve the problem the authors formulate the classification problem as a constrained optimisation problem and introduce a constraint that can be used with existing loss functions to enforce maximal area under the ROC curve (AUC) through prioritizing FPR reduction at high TPR.

**Limitations And Societal Impact:**

Yes

**Main Review:**

The authors propose a new method and they do a got job at explaining it through the usage of a simple example. I really liked the simple example, I think it really helps the reader in understanding what is the objective of the proposed constraints.

Regarding the simple example, I think the cost of the constraint should decrease by $\frac{39\Delta^2}{50}+\frac{3\lambda\Delta}{25}$ in the first case, and by $\frac{19\Delta^2}{50}+ \frac{11 \lambda \Delta}{25}$ in the second case, and not by $39\Delta^2+3\lambda\Delta$ in the first case, and by $19\Delta^2+ 11 \lambda \Delta$ in the second case.

Further the method has been tested on a variety of datasets, with different ratios between the critical and the non-critical class.

Regarding the experiments, I have the following concerns:
- In both the binary case and the multi-class case the distribution of the train set and the validation set are different. Indeed, we have highly imbalanced datasets for the training sets, but balanced datasets for the test sets. Would the results be the same if the authors picked test sets that are as imbalanced as the training set? (which is also what we expect to happen when we apply the method in the real world)
-In the experiments on the MRI dataset, what is the reason of picking just FPR@0FN and FPR@1FN? I think it would have been interesting to show at least FPR@2FN, and maybe FPR@5FN (just to see how the performance varies when allowing for more false negatives)
- In the multi class experiments, why do the authors report the overall accuracy and not the AUC score (maybe averaged in different ways)?
- Is there any reason why the authors did not try MBAUC+ALM?

Regarding the readability, the authors did a good job in explaining the intuitions behind the proposed method, however, the section about the experimental analysis can be confusing:
- $\rho$ is introduced only at page 7, and the reader does not really know where this hyperparameter stems from. I would encourage the authors to include Algorithm 1 (in appendix) to the main paper.
- $ALM_{m,1}$ and $ALM_{m,2}$ are never formally introduced, and they are first mentioned at page 8.
- the authors did not specify what does the bold and underlined in the tables mean. I assume that bold means that the ALM+method is able to perform better than the method alone, while the underline highlights the best method. I would encourage the authors to better explain the above, and also to bold the results of the methods that are able to outperform method+ALM (otherwise the tables can be misleading)
- is there any reason why the Avg AUC in Table 3 has no bold or underlined results?

**After reading authors' comments**: The authors addressed all my concerns. I hope they will include the results showed in the rebuttal phase in the appendix of the final paper. I changed my score from 6 to 7.


**Time Spent Reviewing:**

4

---

> ### Author Response · Authors · 2021-08-10
> **Response to Reviewer 47qX**
>
> **2. Response to comment about including results at higher FNR for MRI dataset.**
>
> As suggested, we present here the  FPR on the MRI dataset @2FN, and @5FN. Overall, it can be observed that the largest benefit from ALM is obtained at higher TPR, consistently with our goal.
>
>
> | Method	| FPR@2FN	| FPR@5FN	|
> | :---	| :----:	| :----:	|
> |		| 		|		|
> | BCE		| 31.3	| 12.5	|
> | S-ML	| 37.5	| 9.4		|
> | S-FL	| 25.0	| 4.5		|
> | A-ML	| 26.6	| 4.5		|
> | A-FL	| 21.8	| 4.5		|
> | CB-BCE	| 23.4	| 1.6		|
> | W-BCE	| 32.8 	| 4.5		|
> | LDAM	| 25.0 	| 4.5		|
> |||
> | ALM+BCE	| **20.3**	| **1.6**		|
> | ALM+S-ML	| **21.8**	| **1.6**		|
> | ALM+S-FL	| **21.5**	| **1.6**		|
> | ALM+A-ML	| **21.8**	| 4.5		|
> | ALM+A-FL	| **15.6**	| **1.6**		|
> | ALM+CB-BCE| **21.5**	| 1.6		|
> | ALM+W-BCE	| **21.8** 	| **1.6**		|
> | ALM+LDAM	| **21.5** 	| **1.6**		|
>
> **3. Response to comment about not using AUC in multi-class experiment.**
>
> For the multi-class setting, we decided to show the overall accuracy instead of the AUC, as the latter is not well-defined in multi-class classification tasks. Moreover, recent works [1,2,3] from the SoA on class imbalance use accuracy to assess the performance on these datasets and we tried to be as consistent as possible with them.
>
> [1] Ren, J., Yu, C., Sheng, S., Ma, X., Zhao, H., Yi, S., Li, H.: Balanced meta-softmax for long- tailed visual recognition. In: Proceedings of Neural Information Processing Systems (NeurIPS) (Dec 2020)
>
> [2] Cao, K., Wei, C., Gaidon, A., Arechiga, N., Ma, T.: Learning imbalanced datasets with label distribution-aware margin loss. In: Wallach, H., Larochelle, H., Beygelzimer, A., d'Alché-Buc, F., Fox, E., Garnett, R. (eds.) Advances in Neural Information Processing Systems. vol. 32. Curran Associates, Inc. (2019)
>
> [3] Cui, Y., Jia, M., Lin, T.Y., Song, Y., Belongie, S.: Class-balanced loss based on effective number of samples. In: Proceedings of the IEEE/CVF Conference on Computer Vision and Pattern Recognition. pp. 9268–9277 (2019)
>
> **4. Response to comment about not trying MBAUC+ALM.**
>
> MBAUC is also an AUC-based optimization approach. We thought it would be redundant to optimize AUC, as both objective function and constraint.
>
> **5. Response to comment regarding the example.**
>
> We thank the reviewer for the correction.
>
> **6. Response to comments on readability.**
>
> We thank the reviewer for all the useful comments and suggestions for a better readability and we will be happy to apply them in the final version if we are allowed.
> In particular, we agree that including Algorithm 1 in the main paper would be beneficial for a better understanding of the proposed method. We opted for this solution due to space reasons and if it is possible, we will move it from supplementary material to the main paper.
>
> Specifically, $\rho$ is a parameter used to regulate the update of the penalty coefficient between epochs.

---

> ### Author Response · Authors · 2021-08-10
> **Response to Reviewer 47qX**
>
> Thank you for the detailed comments and suggestions!
>
> **1. Response to comment about using an imbalanced test set.**
>
> For the in-house medical dataset we provide results on an equally skewed dataset, as it is inherently imbalanced and we show the effectiveness of ALM on it. On the other hand, in this work we decided to test the proposed solution on a balanced test set for CIFAR datasets in order to be consistent with the latest works from the SoA on class imbalance [1, 2, 3].
> We have been encouraged to use this setting by the reviewers of ICML, where a previous version of this paper has been submitted to. In the earlier study, we tested ALM on a smaller version of CIFAR10 with consistent imbalance among training, validation and test sets. However, we were suggested to conform to the common practice of evaluating on a balanced test set, as it is presented in the current version.
> We provide the results obtained from the previous submission in the table below on CIFAR10 at a class ratio of 1:2, 1:9, 1:19.
>
> Class ratio 1:2:
>
> | Method	| FPR@100TPR| FPR@95TPR| FPR@90TPR	|  AUC	|
> | :-------:	|:--------:	| :-------:	| :-------:	| :-------:	|
> |		| 		|		| 		|		|
> | BCE		| 69.1	| 20.9	| 15.9	| 94.7	|
> | S-ML	| 68.2	| 21.6	| 15.2	| 94.5	|
> | S-FL	| 64.3	| 22.0 | 15.6	| 94.7	|
> | A-ML	| 67.0	| 21.8	| 15.6	| 94.5	|
> | A-FL	| 65.1	| 20.1	| 13.8	| 94.8	|
> | CB-BCE	| 69.3	| 23.5	| 16.3	| 94.3	|
> | W-BCE	| 68.7	| 21.9	| 15.6	| 94.5	|
> | MBAUC	|  69.6	| 23.8	| 15.4	| 94.2	|
> |||||
> | ALM+BCE	| **66.2**	| 21.1	| **13.2**	| **95.4**	|
> | ALM+S-ML	| **61.9**	| 21.9	| **13.5**	| **95.1**	|
> | ALM+S-FL	| **54.3**	| **20.5**	| **14.7**	| **95.2**	|
> | ALM+A-ML	| **65.0**	| **21.4**	| **14.3**	| **95.1**	|
> | ALM+A-FL	| **64.0**	| 20.5	| 14.7	| **94.9**	|
> | ALM+CB-BCE| **58.8**	| **18.5**	| **13.4**	| **95.4**	|
> | ALM+W-BCE	|  **59.9**	| **18.7**	| **13.2**	| **95.6**	|
> |||||
>
> Class ratio 1:9:
>
> | Method	| FPR@100TPR| FPR@95TPR| FPR@90TPR	|  AUC	|
> | :-------:	|:--------:	| :-------:	| :-------:	| :-------:	|
> |		| 		|		| 		|		|
> | BCE		| 41.3	| 20.6	| 16.6	| 94.0	|
> | S-ML	| 39.6	| 20.3	| 16.2	| 94.1	|
> | S-FL	| 39.7	| 19.4 	| 17.6	| 93.4	|
> | A-ML	| 42.1	| 21.3	| 17.3	| 93.7	|
> | A-FL	| 42.0	| 20.9	| 17.1	| 93.7	|
> | CB-BCE	| 58.5	| 31.7	| 27.1	| 90.8	|
> | W-BCE	| 54.1	| 30.5	| 23.4	| 91.1	|
> | MBAUC	| 44.1	| 22.7	| 17.0	| 92.1	|
> |||||
> | ALM+BCE	| **34.2**	| **19.0**	| **14.1**	| **94.7**	|
> | ALM+S-ML	| **28.5**	| **20.0**	| **14.0**	| **95.0** 	|
> | ALM+S-FL	| **35.6**	| 22.5	| **17.4**	| **94.2**	|
> | ALM+A-ML	| **32.9**	| **18.9**	| **13.6**	| **94.9**	|
> | ALM+A-FL	| **31.4**	| **16.5**	| **12.4**	| **95.4**	|
> | ALM+CB-BCE| **38.5**	| **22.7**	| **17.9**	| **93.9**	|
> | ALM+W-BCE	| **47.9**	| **23.1**	| **19.8**	| **93.0**	|
> |||||
>
> Class ratio 1:19:
>
> | Method	| FPR@100TPR| FPR@95TPR| FPR@90TPR	|  AUC	|
> | :-------:	|:--------:	| :-------:	| :-------:	| :-------:	|
> |		| 		|		| 		|		|
> | BCE		| 40.4	| 27.9	| 21.1	| 91.9	|
> | S-ML	| 36.4	| 27.9	| 22.0	| 92.2	|
> | S-FL	| 39.3	| 23.5 	| 22.7	| 92.1	|
> | A-ML	| 34.2	| 27.4	| 22.6	| 91.7	|
> | A-FL	| 44.8	| 32.5	| 21.7	| 91.8	|
> | CB-BCE	| 61.6	| 36.5	| 35.1	| 88.2	|
> | W-BCE	|  61.2	| 41.2	| 32.7	| 88.8	|
> | MBAUC	| 36.0	| 26.4	| 25.0	| 91.8	|
> |||||
> | ALM+BCE	| **29.8**	| **27.5**	| **17.2**	| **93.2**	|
> | ALM+S-ML	| **28.4**	| **24.1**	| **17.9**	| **93.7** 	|
> | ALM+S-FL	| **31.0**	| 23.5	| **16.8**	| **93.5**	|
> | ALM+A-ML	| **29.0**	| **26.2**	| **22.4**	| **93.0**	|
> | ALM+A-FL	| **34.4**	| **27.7**	| 22.3	| **93.4**	|
> | ALM+CB-BCE| **52.6**	| 42.1 |  **25.7**	| **90.8**	|
> | ALM+W-BCE	| **48.2**	| **31.2**	| **19.6**	| **91.2**	|
> |||||
>
> [1] Ren, J., Yu, C., Sheng, S., Ma, X., Zhao, H., Yi, S., Li, H.: Balanced meta-softmax for long- tailed visual recognition. In: Proceedings of Neural Information Processing Systems (NeurIPS) (Dec 2020)
>
> [2] Cao, K., Wei, C., Gaidon, A., Arechiga, N., Ma, T.: Learning imbalanced datasets with label distribution-aware margin loss. In: Wallach, H., Larochelle, H., Beygelzimer, A., d'Alché-Buc, F., Fox, E., Garnett, R. (eds.) Advances in Neural Information Processing Systems. vol. 32. Curran Associates, Inc. (2019)
>
> [3] Cui, Y., Jia, M., Lin, T.Y., Song, Y., Belongie, S.: Class-balanced loss based on effective number of samples. In: Proceedings of the IEEE/CVF Conference on Computer Vision and Pattern Recognition. pp. 9268–9277 (2019)

---

### Official Review · Reviewer_w5h3 · 2021-07-16

**Rating:** 7
**Confidence:** 4

**Summary:**

This paper proposes a very interesting approach for training DNNs on binary (and multiclass) classification tasks with critical under-represented classes, which is very typical in domains like healthcare. In this approach, the task is posed as a constrained optimization problem and a novel constraint is designed that emphasizes on reducing the false positive rate while maintaining a high true positive rate, which is then added to the loss function and solved using an Augmented Lagrangian Method. The main contribution of the paper is proposing a novel constraint optimization approach that can be added to any previous work, and that improves upon the state of the art results.

**Limitations And Societal Impact:**

- One main limitation that the authors discussed is that they performed their experiments on the small MRI dataset, which can make the results unreliable. They addressed this by using an ensembling strategy.
- There is no obvious potential negative societal impact of this work. The authors further mention that fairness is one of the motivations of this work.

**Main Review:**

Originality:
- The proposed approach is definitely very novel and interesting. I am not aware of other work in this area that pose AUC optimization as a constrained optimization problem.
- The authors provided a very comprehensive and well organized related work section, and its clear how this approach distinguishes itself from the others. The related work is broadly split into 3 groups: the first proposes variants of common loss functions with added emphasis on minority classes, the second focuses on different sampling strategies to improve the occurrences of the minority class samples in the batches, and the third focuses on scaling and thresholding model outputs post training. In the first group, another subgroup of papers focus on AUC optimization. This paper falls in that subgroup, but differs from the other papers by proposing an approach that's scalable to large datasets and model architectures, and that can be used along with other loss functions.

Clarity:
- The paper overall is very well written. Its clear, understandable, and very well organized. The code is also provided in the supplementary material, and the algorithms are very well explained in the paper.
- It was very helpful that the authors included a toy example in section 4.2 to explain intuitively how their approach works.
- One minor comment is, in Figure 1, the axis labels should be False Positive Rate and True Positive Rate instead.

Quality:
- The paper is technically very sound. The proposed constraint was given ample intuitive and mathematical explanation, and using a toy example in section 4.2 and detailed theoritical explanations in the appendix section A.8 is very helpful.
- The authors conducted very comprehensive experiments (including the results presented in the appendix) and provided compelling evidence for the improvements obtained by their approach by comparing directly with the results of several approaches from the related work. This approach improved the results considerably in most of the cases.
- The authors also extended their work to the multiclass domain and provided the relevant results.

Significance:
- The proposed approach produced considerable improvements in empirical results over several state of the art methods. The main appeal of this approach is that it can be integrated with any other approach very easily. I think this will make it very easy for other authors to include it as a baseline in their works. This is a complete piece of work with a several important details provided in the appendix, and code required to reproduce the results provided in the supplementary material.
- The proposed approach provides a unique way to reduce the FPRs of classifiers operating at a high TPR. This is a very wide spread problem faced especially in the medical domain, and a lot of approaches do lead to a high FPR which impacts their practical usability adversely. This approach is definitely a very welcome addition in tackling that problem.

Weaknesses:
- It would have been helpful if the authors provided results in another public medical dataset like ADNI, etc.
- It would be helpful if the authors include some discussion on how they plan to extend this work in the conclusion section.


**Time Spent Reviewing:**

5

---

> ### Author Response · Authors · 2021-08-10
> **Response to Reviewer w5h3**
>
> Thank you for the detailed review and the positive comments!
>
> **1. Possible extensions for the future.**
>
> As class imbalance can be a severe and inherent problem in segmentation as well, one possible extension for the future could be applying the proposed work to such a task, especially for those applications where under-segmentation can have severe consequences, e.g. tumor segmentation. Although the current method is not directly applicable, a similar problem is also present in image reconstruction from partial observations, where clinically relevant parts of an image, e.g. lesions, must have a higher fidelity than other regions. Moreover, we agree that it would be interesting to evaluate the proposed method on new medical datasets, such as ADNI.

---

### Official Review · Reviewer_hCqm · 2021-07-31

**Rating:** 6
**Confidence:** 4

**Summary:**

Summary
This paper proposes a new optimization approach to classification problems with class imbalance, where correctly classifying the minority class is more important than correctly classifying the majority class. One specific motivation is in medical imaging diagnostics, where misclassifying the “critical” minority class (e.g., diseased) is a more severe mistake than misclassifying the majority class (e.g., healthy). The authors therefore use an augmented Lagrangian method (ALM) approach to minimize the FPR for the majority while maintaining a high TPR for the critical class in a binary classification problem, as well as suggesting an optimization formulation for multi-class classification tasks. The ALM approach tries to enforce a margin between the critical and non-critical samples in the output space, and is structured to penalize a critical misclassification more than a non-critical misclassification.

The problem is certainly important for medical diagnostic tools, as well as other problems where errors have different costs for different classes. The proposed method is a useful approach for these problems, and the theoretical analysis of the asymmetry and required imbalance of the problem is clear and interesting. The experiments show some benefit of the ALM-based approach and its ability to be coupled with many different loss functions. The writing is clear overall, but with various grammatical mistakes throughout (and particularly in the supplement). My major concerns are the significance of the result and the setting of the CIFAR experiments.


**Limitations And Societal Impact:**

I don't have any concern about the negative societal impact of this work.

**Main Review:**

Major Comments
1. The paper does not compare with any AUC-optimization-based approaches apart from MBAUC. In Section 2, the authors say that “... all aforementioned AUC optimization-based methods were applied to linear predictive models and their performance on DNNs are unknown”. Why not compare against any of these approaches? Or is it not just that the performance is unknown, but that their application to DNNs is nontrivial?
2. Significance of the result. Statistical significance is not included in Table 1 and Table 2. Multiple replicates should be used to test the significance. Some numbers are really close in table 1,2 and still marked in bold text (better performance). Avg AUCs in table 3 have large standard deviation, which might not indicate significant improvement. Instead of using selected thresholds 98%, 95% and 90%, it is more informative to show a TPR- FPR curve plot.
3. How to predefine critical and non-critical classes? In binary classification, it seems that the minority class is always the critical class. Is this always the case in real-world applications? In the multi-class setting with many classes, classifying each class into critical and non-critical seems to be difficult and needs domain knowledge. If a critical class is misdefined as a non-critical class, this might be problematic for many applications.
4. Setting of class ratios in CIFAR experiments. Since this paper is motivated by clinical applications, the prostate cancer example makes sense to me. However, in the CIFAR experiments, very large ratios of 1:100 and 1:200 are used. This ratio seems to be unrealistic (e.g., a clinical dataset with 200 healthy patients and 1 sick patient). Is it because the method can only have good improvement on such a very imbalanced ratio? I suggest the author to either show improvement on CIFAR with smaller ratios (e.g., 1:5, 1:10) or find a real-world medical application that has such a class ratio close to 1:100.

Minor Comments
1. A slightly longer description of MBAUC, such as the framework it uses or other ways in which it differs from the proposed method, would be useful.
2. In Section 3, the authors say that other methods “...suffer from training instability and convergence”. I think you mean non-convergence?
It would be useful to include a citation for the claim in Section 4.1 that “Satisfying the constraint would directly ensure maximal AUC.”
3. In Figure 4.2, given the precise setup of the line plot I think that it would be useful for the grid axes to go from [0, 5], rather than [0, 1].
4. In Section 5.1, what is the justification for CIFAR10 and CIFAR100 to use validation sets that have 100 samples per class? Why is this not also class-imbalanced?
5. I find the tables difficult to read. First, when the original loss function achieves a lower FPR / higher test AUC than the ALM-optimization version, I think those results should be bolded for clarity. Given the quantity of bolded results, it is also difficult to distinguish the underlines of “best result”, particularly when it is not one of the bolded entries. Also, given my understanding of what “bold” means in these tables, I didn’t understand why the entry for ALM_{m, 1} + LDAM [Acc. Critical Class 1] was not bolded in table A5.
6. The details of the multi-class classification experiment were not fully explained in the paper. Specifically, I was not fully clear that there was only one critical class (and multiple non-critical classes) in this problem formulation until Section A.4.
7. I wish the authors provided slightly more discussion about the actual results, including why the ALM helped more with some methods than others. One particularly interesting result for me was the performance of BCE, which in theory does not take into account class imbalance, but was one of the best-performing baselines and, in some cases, outperformed or was very close to ALM + BCE (particularly on the CIFAR data). This wasn’t the case on the MRI dataset -- is this due to an easier problem (lower FPR/higher TPR to begin with)? Weaker requirements on TPR? Less extreme class imbalances?
8. I would appreciate some kind of study on the impact of class imbalance ratios to the improvement attributed to the ALM optimization. This would help me understand more details about the kind of problem that the ALM optimization is most suited to, as well as potentially answer some of the questions in the point above.
9. Showing the % improvement by incorporating ALM would also be a useful plot to include somewhere, and might be more readable than the tables.
10. Tables A.3 and A.4 have less convincing improvements than the binary classification task (and particularly table A.4). A little more discussion here to connect these results with Table A.5 would be useful. One specific confusion with table A.5 was the low critical class accuracy -- given the paper’s motivation, could you increase the required TPR? Currently there is much lower accuracy on the critical class than the non-critical classes, which seems counter to the main goal. Is this mitigated at all with a less extreme class imbalance ratio?
In Figure A.2, what is Delta_N set to? Given the prior discussion about Delta_P and Delta_N, this would be useful. Also, is Delta_E just the Delta in the text?
11. The tables (and table numbers) seem to be missing from Section A.10.
12. The most consistent grammar mistake was “... critical class’ ….” The singular possessive of “class” is “class’s.”

Update -- I raised my score after reading the response from authors. Thank you for the detailed explanation and additional results.


**Time Spent Reviewing:**

3

---

> ### Author Response · Authors · 2021-08-10
> **Response to Reviewer hCqm**
>
> **3. Response to Major comment 1: Other AUC optimization based approaches other than MBAUC are nontrivial to apply in DNNs.**
>
> AUC optimization is based on optimizing Mann-Whitney statistics which is an NP-hard problem [1]. Therefore, one needs to make some approximations for AUC optimization. Most of the existing AUC optimization based methods simplify Mann-Whitney statistics by assuming that the model is linear. Therefore, it is non-trivial to apply these methods in a non-linear setting with DNNs. To the best of our knowledge, the only method that performs AUC optimization in a non-linear setting is MBAUC. Therefore, we compare the proposed approach only with MBAUC in our experiments.
>
> [1] Mann, H.B., Whitney, D.R.: On a test of whether one of two random variables is stochastically larger than the other. The annals of mathematical statistics pp. 50–60 (1947)
>
> **4. Response to Major comment 3: selection of critical classes.**
>
> **4.1 How to predefine critical and non-critical classes?**
>
> We define the critical classes as the ones that have fewer samples and higher misclassification costs compared to others. All the other classes are defined as non-critical.
>
> **4.2 In binary classification, it seems that the minority class is always the critical class. Is this always the case in real-world applications?**
>
> Our motivation comes from a medical imaging application where misclassification cost for minority class samples is higher than the majority class samples. This situation arises in applications in various disciplines, including oncology and cardiology, which form our main motivation. Moreover, networks tend to make more mistakes for the minority class samples since naive training biases them toward majority classes due to class imbalance. Therefore, in this paper, we always chose critical classes as the minority classes.
>
> One can find a real application where making mistakes for the majority class is more critical. However, in this setting, networks are less likely to make severe mistakes and we believe our setting where critical classes are under-represented is a more common problem in medical applications and, arguably, a harder one.
>
> **4.3 In the multi-class setting with many classes, classifying each class into critical and non-critical seems to be difficult and needs domain knowledge. If a critical class is misdefined as a non-critical class, this might be problematic for many applications.**
>
> We agree that predefining critical and non-critical classes is very crucial. This needs to be done with domain knowledge since critical classes can be best determined by domain experts in real applications. It is indeed our motivation to work towards solutions with domain experts who can identify the critical classes and mistakes with severe consequences.
>
> **5. Response to Minor comment 1: differences with MBAUC.**
> The difference between MBAUC and the proposed work lies in the way the notion of AUC is incorporated into the loss function. MBAUC aims at optimizing AUC directly, while ALM applies it as a constraint to minimize the FPRs at maximal TPRs. This allows combining ALM with other types of losses and ends up being a more effective approach.  Moreover, the goal of MBAUC is to optimize the overall AUC, while the aim of this work is to exploit the definition of AUC to improve the performance at high TPR.
>
> **6. Response to Minor comment 2.**
> We thank the reviewer for the correction, we meant “non-convergence”. Regarding the required reference, we would put [1].
>
> [1] Mann, H.B., Whitney, D.R.: On a test of whether one of two random variables is stochastically larger than the other. The annals of mathematical statistics pp. 50–60 (1947)
>
> **7. Response to Minor comment 3: Figure 1.**
> In Figure 1, the axes are meant to represent TPR and FPR, we will change their labels accordingly.
>
> **8. Response to Minor comment 4: Validation set for CIFAR10 and CIFAR100.**
> The number of validation samples is selected arbitrarily for each dataset, so that we have a reasonable number of images to carry out validation. We chose to use a balanced validation and test set to be consistent with previous works and SoA on class imbalance [1,2,3]. This practice is done to analyze the impact of an imbalanced training set on a balanced test set. In contrast, the in-house medical dataset has inherently imbalance both in training and test set.
>
> [1] Ren, J., Yu, C., Sheng, S., Ma, X., Zhao, H., Yi, S., Li, H.: Balanced meta-softmax for long- tailed visual recognition. In: Proceedings of Neural Information Processing Systems (NeurIPS) (Dec 2020)
>
> [2] Cao, K., Wei, C., Gaidon, A., Arechiga, N., Ma, T.: Learning imbalanced datasets with label distribution-aware margin loss. In: Wallach, H., Larochelle, H., Beygelzimer, A., d'Alché-Buc, F., Fox, E., Garnett, R. (eds.) Advances in Neural Information Processing Systems. vol. 32. Curran Associates, Inc. (2019)
>
> [3] Cui, Y., Jia, M., Lin, T.Y., Song, Y., Belongie, S.: Class-balanced loss based on effective number of samples. In: Proceedings of the IEEE/CVF Conference on Computer Vision and Pattern Recognition. pp. 9268–9277 (2019)
>
> **9. Response to Minor comment 7: additional discussion about results.**
> According to the results in the paper, we observe that ALM improves more when the baseline is performing worse. We calculated Pearson correlation between the average baseline AUC vs average improvement (difference of AUC between baseline and ALM) and obtained -0.85 which indicates a very high negative correlation.
>
> We agree with the reviewer that BCE is performing quite well compared to some other baselines (e.g. CB-BCE, W-BCE) on CIFAR datasets. One possible reason could be that the classification problems on CIFAR10 and CIFAR100 are relatively easy compared to MRI dataset such that BCE can handle well despite class imbalance. This is confirmed by the results obtained for the standard deviation of AUC over the 10 different runs, whose results are reported in Tables 4, A.6 and A.7. The std is consistently larger for the MRI dataset, compared to CIFAR10 and CIFAR100, reflecting a higher uncertainty of the network on the predictions [2].
>
> [2] Lakshminarayanan, Balaji & Pritzel, Alexander & Blundell, Charles. Simple and Scalable Predictive Uncertainty Estimation using Deep Ensembles. Advances in Neural Information Processing Systems (NIPS 2017)
>
> **10. Response to Minor comment 8: ALM and level of imbalance.**
> On CIFAR10, we performed experiments with different class imbalance ratios of 1:50, 1:100, 1:200 (see Table 1 and A1.1). We observe that both baseline and ALM accuracy diminish as the class imbalance rate increases which is expected. However, we don't observe any correlation between the improvement that ALM achieves and class imbalance ratio. We delved into the ALM loss function to understand this behaviour. There are two factors that determine the contributions of Lagrangian and penalty terms given that number of negative class examples ($n$) is fixed for different class imbalance ratios: number of positive examples ($p$) and difference between negative and positive pairs ($d = f(x_j^p) - f(x_k^n)$) (see Eq. 4). At high class imbalance ratios, $p$ reduces. Although, d may be affected by $p$, it may also be affected by other factors e.g. dataset. So, it is difficult to draw a conclusion of ALM as a function of $p$ which determines the class imbalance ratio when $n$ is fixed.
>
> **11. Response to Minor comment 9: include the % improvement.**
> We thank the reviewer for the suggestion. If the rules allow us, we will include such plots in supplementary material.
>
> **12.  Response to Minor comment 10: clarifications about Tables A.3, A.4 and Fig. A.2.**
>
> **12.1** As shown in Tables A.3 and A.4, due to the extreme class imbalance and the increased complexity of multi-class classification, the accuracy in the minority class generally decreases. However, the goal here is not to show that critical classes have higher accuracy than their non-critical counterparts. Our objective is to show that by applying the proposed constraint to the baseline and solving the problem with an ALM, the accuracy on the critical class increases in almost all cases compared to baselines, without significantly degrading the accuracy on all other classes (in the rare cases where accuracy on non-critical classes decrease, the degradation on all other 8 classes is always less than 0.5%).
>
> **12.2** Our aim in Figure A.2 and the related section is to give some theoretical insight on how the proposed constraint encourages high TPR at low FPR. Therefore, we show a visual example and use generic notations for distances between samples instead of specific values to drive the theoretical insight in the section. Consequently, in this figure, neither $\Delta_N$  nor $\Delta$ and $\Delta_P$ are not set to a specific value. At the end of the discussion in the section, we demonstrate that $\Delta_N$ should be larger than $\Delta_P$ by a certain limit $\Delta_{diff,lim}$ to enforce the same loss to both negative and positive samples. In other words, these analyses indicate that the proposed loss assigns higher loss for misclassified positive (minority) class samples than the negative ones as long as $\Delta_N \leq \Delta_P + \Delta_{diff,lim}$ which encourages achieving high TPR and low FPR.

---

> ### Author Response · Authors · 2021-08-10
> **Response to Reviewer hCqm**
>
>
>
> **2. Response to Major comment 4: settings of CIFAR experiments.**
>
> In this work we decided to test the proposed solution on CIFAR datasets under such extreme imbalances in order to be consistent with the latest works from the SoA on class imbalance [1,2,3].
> We have also been encouraged to use this setting by the reviewers of ICML conference, where a previous version of this paper has been submitted to. In the earlier version of this work, we tested ALM on a smaller version of CIFAR10 and injected class imbalance with ratios 1:2, 1:9, and 1:19. However, we were suggested to conform to the common benchmark, which is adopted in the current version.
> We provide the results obtained from the previous submission in the table below on CIFAR10 at a class ratio of 1:2, 1:9, 1:19.
>
> Class ratio 1:2:
>
> | Method	| FPR@100TPR| FPR@95TPR| FPR@90TPR	|  AUC	|
> | :-------:	|:--------:	| :-------:	| :-------:	| :-------:	|
> |		| 		|		| 		|		|
> | BCE		| 69.1	| 20.9	| 15.9	| 94.7	|
> | S-ML	| 68.2	| 21.6	| 15.2	| 94.5	|
> | S-FL	| 64.3	| 22.0 | 15.6	| 94.7	|
> | A-ML	| 67.0	| 21.8	| 15.6	| 94.5	|
> | A-FL	| 65.1	| 20.1	| 13.8	| 94.8	|
> | CB-BCE	| 69.3	| 23.5	| 16.3	| 94.3	|
> | W-BCE	| 68.7	| 21.9	| 15.6	| 94.5	|
> | MBAUC	|  69.6	| 23.8	| 15.4	| 94.2	|
> |||||
> | ALM+BCE	| **66.2**	| 21.1	| **13.2**	| **95.4**	|
> | ALM+S-ML	| **61.9**	| 21.9	| **13.5**	| **95.1**	|
> | ALM+S-FL	| **54.3**	| **20.5**	| **14.7**	| **95.2**	|
> | ALM+A-ML	| **65.0**	| **21.4**	| **14.3**	| **95.1**	|
> | ALM+A-FL	| **64.0**	| 20.5	| 14.7	| **94.9**	|
> | ALM+CB-BCE| **58.8**	| **18.5**	| **13.4**	| **95.4**	|
> | ALM+W-BCE	|  **59.9**	| **18.7**	| **13.2**	| **95.6**	|
> |||||
>
> Class ratio 1:9:
>
> | Method	| FPR@100TPR| FPR@95TPR| FPR@90TPR	|  AUC	|
> | :-------:	|:--------:	| :-------:	| :-------:	| :-------:	|
> |		| 		|		| 		|		|
> | BCE		| 41.3	| 20.6	| 16.6	| 94.0	|
> | S-ML	| 39.6	| 20.3	| 16.2	| 94.1	|
> | S-FL	| 39.7	| 19.4 	| 17.6	| 93.4	|
> | A-ML	| 42.1	| 21.3	| 17.3	| 93.7	|
> | A-FL	| 42.0	| 20.9	| 17.1	| 93.7	|
> | CB-BCE	| 58.5	| 31.7	| 27.1	| 90.8	|
> | W-BCE	| 54.1	| 30.5	| 23.4	| 91.1	|
> | MBAUC	| 44.1	| 22.7	| 17.0	| 92.1	|
> |||||
> | ALM+BCE	| **34.2**	| **19.0**	| **14.1**	| **94.7**	|
> | ALM+S-ML	| **28.5**	| **20.0**	| **14.0**	| **95.0** 	|
> | ALM+S-FL	| **35.6**	| 22.5	| **17.4**	| **94.2**	|
> | ALM+A-ML	| **32.9**	| **18.9**	| **13.6**	| **94.9**	|
> | ALM+A-FL	| **31.4**	| **16.5**	| **12.4**	| **95.4**	|
> | ALM+CB-BCE| **38.5**	| **22.7**	| **17.9**	| **93.9**	|
> | ALM+W-BCE	| **47.9**	| **23.1**	| **19.8**	| **93.0**	|
> |||||
>
> Class ratio 1:19:
>
> | Method	| FPR@100TPR| FPR@95TPR| FPR@90TPR	|  AUC	|
> | :-------:	|:--------:	| :-------:	| :-------:	| :-------:	|
> |		| 		|		| 		|		|
> | BCE		| 40.4	| 27.9	| 21.1	| 91.9	|
> | S-ML	| 36.4	| 27.9	| 22.0	| 92.2	|
> | S-FL	| 39.3	| 23.5 	| 22.7	| 92.1	|
> | A-ML	| 34.2	| 27.4	| 22.6	| 91.7	|
> | A-FL	| 44.8	| 32.5	| 21.7	| 91.8	|
> | CB-BCE	| 61.6	| 36.5	| 35.1	| 88.2	|
> | W-BCE	|  61.2	| 41.2	| 32.7	| 88.8	|
> | MBAUC	| 36.0	| 26.4	| 25.0	| 91.8	|
> |||||
> | ALM+BCE	| **29.8**	| **27.5**	| **17.2**	| **93.2**	|
> | ALM+S-ML	| **28.4**	| **24.1**	| **17.9**	| **93.7** 	|
> | ALM+S-FL	| **31.0**	| 23.5	| **16.8**	| **93.5**	|
> | ALM+A-ML	| **29.0**	| **26.2**	| **22.4**	| **93.0**	|
> | ALM+A-FL	| **34.4**	| **27.7**	| 22.3	| **93.4**	|
> | ALM+CB-BCE| **52.6**	| 42.1 |  **25.7**	| **90.8**	|
> | ALM+W-BCE	| **48.2**	| **31.2**	| **19.6**	| **91.2**	|
> |||||
>
> [1] Ren, J., Yu, C., Sheng, S., Ma, X., Zhao, H., Yi, S., Li, H.: Balanced meta-softmax for long- tailed visual recognition. In: Proceedings of Neural Information Processing Systems (NeurIPS) (Dec 2020)
>
> [2] Cao, K., Wei, C., Gaidon, A., Arechiga, N., Ma, T.: Learning imbalanced datasets with label distribution-aware margin loss. In: Wallach, H., Larochelle, H., Beygelzimer, A., d'Alché-Buc, F., Fox, E., Garnett, R. (eds.) Advances in Neural Information Processing Systems. vol. 32. Curran Associates, Inc. (2019)
>
> [3] Cui, Y., Jia, M., Lin, T.Y., Song, Y., Belongie, S.: Class-balanced loss based on effective number of samples. In: Proceedings of the IEEE/CVF Conference on Computer Vision and Pattern Recognition. pp. 9268–9277 (2019)

---

> ### Author Response · Authors · 2021-08-10
> **Response to Reviewer hCqm**
>
> Thank you for the detailed comments and suggestions!
>
> **1. Response to Major comment 2: Statistical test of multiple runs, TPR-FPR plot.**
>
> **1.1.** All the results presented in the paper are obtained by averaging 10 runs with different random seeds for the model parameters. As suggested by the reviewer, we performed statistical significance analysis on the AUC results using the DeLong test. We copied the results of ALM below from Tables 1, 2, and 3 in the main paper and marked the ones that passes the DeLong test ($p \leq 0.5$) using *. Also, note that we wrote the results where ALM improves baseline using bold font. Therefore, a bold result marked with * indicates that the improvement achieved by ALM over the baseline is statistically significant which is the case in the majority of the cases shown in the tables below.
>
> ALM results from  Table 1 in the main paper
>
> | Method	| CIFAR10 1:100 	| CIFAR10 1:200	|
> | :---	| :----:   		| :---:		|
> |		| AUC			| AUC			|
> | ALM+BCE	| **93.1** * 		| 86.7* 		|
> | ALM+S-ML	|**92.5** *		| **87.9** *		|
> | ALM+S-FL	| 91.5		| **86.9** * 		|
> | ALM+A-ML	| **92.8**		| **87.6** 		|
> | ALM+A-FL	| **92.7** *		| **87.0** *		|
> | ALM+CB-BCE| **88.1** *		| **80.0** *		|
> | ALM+W-BCE	| **89.3** * 		| **81.0** *		|
> | ALM+LDAM	| **91.0** * 		| 85.6*		|
>
>
> ALM results from  Table 2 in the main paper
>
> | Method	| CIFAR100 1:100 	| CIFAR100 1:200	|
> | :---	| :----:   		| :---:		|
> |		| AUC			| AUC			|
> | ALM+BCE	| **82.7** * 		| **80.9** * 		|
> | ALM+S-ML	| 81.7 	      | **80.7** *		|
> | ALM+S-FL	| 81.7 		| **80.8** * 		|
> | ALM+A-ML	| **82.7** *		| **81.0** *		|
> | ALM+A-FL	| **83.2** *		| **80.7** 		|
> | ALM+CB-BCE| **83.8** *		| **81.0** *		|
> | ALM+W-BCE	| **83.2**		| **81.3** *		|
> | ALM+LDAM	| **83.2**		| 81.5		|
>
>
> ALM results from  Table 3 in the main paper
>
> | Method	| MRI		|
> | :---	| :----:	|
> |		| AUC ens.	|
> | ALM+BCE	| **85.4** *	|
> | ALM+S-ML	| **80.3** *	|
> | ALM+S-FL	| **84.2** *	|
> | ALM+A-ML	| **76.4** *	|
> | ALM+A-FL	| **81.5** 	|
> | ALM+CB-BCE| **79.5** 	|
> | ALM+W-BCE	|**81.4** * 	|
> | ALM+LDAM	| **77.0** * 	|
>
> **1.2.** We agree that TPR-FPR plots would be more informative than using some thresholds. However, presenting TPR-FPR plots for each baseline and ALM pair (e.g. BCE and ALM+BCE) is not space efficient. And, putting all pairs into the same plot would result in cluttered plots which are hard to interpret. Therefore, we opted for tables showing high levels of TPR. Should we be allowed to do so, we will add some of them to the supplementary materials as illustrative examples.

---

### Decision · Program_Chairs · 2021-09-27

**Decision:**

Accept (Poster)

**Comment:**

This paper proposes a novel approach to train deep neural networks for classification tasks that involve high class imbalance, which is ubiquitous in medical problems.  The novelty of the proposed method is very high, and related works are presented comprehensively.  The paper is overall very well written and structured.  The potential impact of the proposed algorithm is high given that class imbalance is a common problem in many real-world applications.  Reviewers raised major concerns regarding various aspects of experiments, which were successfully addressed by the authors. Overall, this paper constitutes an important contribution to the field and passes the bar for the acceptance to NeurIPS.